# Distinct population codes for attention in the absence and presence of visual stimulation

Adam C. Snyder [1,2,3], Byron M. Yu [1,3,4] & Matthew A. Smith [2,3,5]

Visual neurons respond more vigorously to an attended stimulus than an unattended one. How the brain prepares for response gain in anticipation of that stimulus is not well understood. One prominent proposal is that anticipation is characterized by gain-like modulations of spontaneous activity similar to gains in stimulus responses. Here we test an alternative idea: anticipation is characterized by a mixture of both increases and decreases of spontaneous firing rates. Such a strategy would be adaptive as it supports a simple linear scheme for disentangling internal, modulatory signals from external, sensory inputs. We recorded populations of V4 neurons in monkeys performing an attention task, and found that attention states are signaled by different mixtures of neurons across the population in the presence or absence of a stimulus. Our findings support a move from a stimulation-invariant account of anticipation towards a richer view of attentional modulation in a diverse neuronal population.

---

[1] Department of Electrical and Computer Engineering, Carnegie Mellon University, Pittsburgh 15289 PA, USA. [2] Department of Ophthalmology, University of Pittsburgh, Pittsburgh 15213 PA, USA. [3] Center for the Neural Basis of Cognition, Carnegie Mellon University and University of Pittsburgh, Pittsburgh 15260 PA, USA. [4] Department of Biomedical Engineering, Carnegie Mellon University, Pittsburgh 15289 PA, USA. [5] Department of Bioengineering, University of Pittsburgh, Pittsburgh 15213 PA, USA. These authors contributed equally: Byron M. Yu, Matthew A. Smith. Correspondence and requests for materials should be addressed to M.A.S. (email: matt@smithlab.net)

The ability to select goal-relevant information from the environment while ignoring distractions is essential to successful daily living. Much of the research into the neurophysiological underpinnings of this important cognitive function, known as selective attention, has focused on modulations of stimulus processing. However, attention states are not established instantaneously: they take hundreds of milliseconds to be prepared prior to the arrival of an anticipated stimulus[1,2]. The state of anticipation can then be maintained, notwithstanding occasional lapses, until the expected event[3–5]. We sought to understand how the brain organizes spiking activity to best detect and respond to anticipated sensory stimuli.

The canonical attention effect in visual cortex is that neurons fire more vigorously to an attended stimulus compared to an unattended stimulus[6], and it has been hypothesized that such response gain mechanisms (or, more properly, a normalized response gain[7,8]) might underlie behavioral advantages of attention by amplifying the representation of the relevant information[9,10]. However, such response gains typically appear after 150 to 300 ms following stimulus onset[6,11,12], which is likely too sluggish to fully account for behavioral improvements. Thus, we focused on the neural correlates of attention during earlier time-points, in particular the spontaneous neural activity preceding an anticipated stimulus.

The current prevailing view of anticipatory attention preceding stimulus onset is that it is characterized by the same gain-based mechanisms used to describe attentional modulations of stimulus-evoked responses[13–17]. However, evidence for this stimulation-invariant interpretation of attention has been inconsistent. Some studies have reported slight increases in average baseline firing rates with attention[13–16,18], but others have not[19–25]. Moreover, the reported gains in baseline firing rates are far weaker than corresponding baseline attention effects measured with functional magnetic resonance imaging (fMRI)[26,27] and electroencephalography/magnetoencephalography (EEG/MEG)[28], suggesting that baseline gains in average firing rates may be just the tip of the iceberg.

From first principles, there are reasons to suppose that the neural correlates of attention states might be qualitatively different in the absence, versus presence, of sensory stimulation. Primarily, it would be adaptive not to confuse the internal attention signal with a spurious representation of a stimulus. Consider a simplified example: a hypothetical downstream brain area that determines stimulus strength as a weighted sum of activity across the neural population of interest. For a sensory population these weights would by definition be predominantly positive. If anticipatory attention were simply to increase the spontaneous firing rates of all neurons, as with a stimulation-invariant gain model, then that gain would change the value of the readout for the downstream area, potentially leading to confusing the anticipatory state change as the appearance of a weakly effective stimulus.

As an alternative to a stimulation-invariant gain model, we supposed a mixture of firing rate modulations including both suppression and facilitation would provide a better strategy for anticipatory attention. Then, the average attention-related firing rate increase across our population of interest would be minimal, and the downstream area would be less likely to confuse this state change for a stimulus. In this framework, effects of anticipatory attention would be explicitly obscured by averaging across neurons in a population, but could be uncovered by a separate readout consisting of a different mixture of neural responses where some neurons take on negative weights. This would support a straightforward linear decoding scheme for effectively disentangling internal and external signals.

To test the idea that a mixture of suppression and facilitation characterizes anticipatory attention, we recorded neural populations in visual cortical area V4 of monkeys performing a spatial attention task. The population patterns of attentional firing rate modulations we observed prior to stimulus onset were fundamentally different than the patterns we saw during stimulus processing. Moreover, the distinct features of anticipatory states were predictive of the subjects' behavioral performance. These results defy an interpretation of anticipatory attention based on a stimulation-invariant gain modulation, and indicate the need to reconceptualize the neurophysiological mechanisms underlying the dynamic allocation of attention.

## Results

**Behavioral effects of attention.** We trained two adult male rhesus macaque monkeys (*Macaca mulatta*) to perform a demanding orientation change-detection task (Fig. 1a) in which one of two stimulus locations was block-cued to be more likely to change (the valid target location). The subjects were more accurate (Fig. 1b) and faster (Fig. 1b, insets) at detecting orientation changes at the valid target location compared to the invalid target location, confirming that the subjects selectively attended to the valid location. Our goal was to characterize the changes in V4 neural activity that underlie these behavioral effects.

As an alternative to a stimulation-invariant gain model, we reasoned that a mixture of suppression and facilitation would provide a better strategy for anticipatory attention in the absence of a stimulus. Our analysis approach was twofold: first, we examined the overall spiking response by averaging across the neurons in the population and found that this signal differentiated attention states late after stimulus onset (200–400 ms; the post-stimulus interval), but did not differentiate attention states pre-stimulus (−200 to 0 ms); second, we employed a population-level analysis that allowed for a trial-by-trial estimate of attention involving both increases and decreases in firing rate across the population, which revealed reliable differentiation of anticipatory attention states.

**Slow-latency modulations of population average firing rates.** A typical finding is that spatial attention leads to an average gain of firing rates in V4 of 5–30% for attended compared to unattended stimuli that manifests >150 ms after stimulus onset[6]. Also, typically, little to no modulation is seen during anticipatory periods after attention cues have been given but no imperative stimulus has yet appeared[20,24,29,30]. When we analyzed the average population response, we replicated this combination of results characteristic of the literature (Fig. 2). Firing rates were significantly greater in response to stimuli when the attention cue had been presented in the receptive field (RF) area of the population compared to when the attention cue had been presented in the opposite hemifield. However, the time-course of this modulation was relatively slow: the difference was statistically detectable starting at 233 ms, which was slower than the average median saccadic response time for the task (mean ± SD: 199.5 ± 9.1 ms for Monkey W; 173.2 ± 4.0 ms for Monkey P; Fig. 2 inset). Even recognizing that the timing of statistical significance varies due to assumed significance level and other factors, such a slow time course of attention effects presents a conundrum because the neural effects of attention must strictly precede the behavioral benefits. Moreover, the saccadic response time is undoubtedly an overestimate of the subjects' true decision times: monkeys' stop-signal reaction times (an estimate of internal decision processes) have been measured to be about 110 ms in a task where the average saccadic reaction times are greater than 250 ms[31,32].

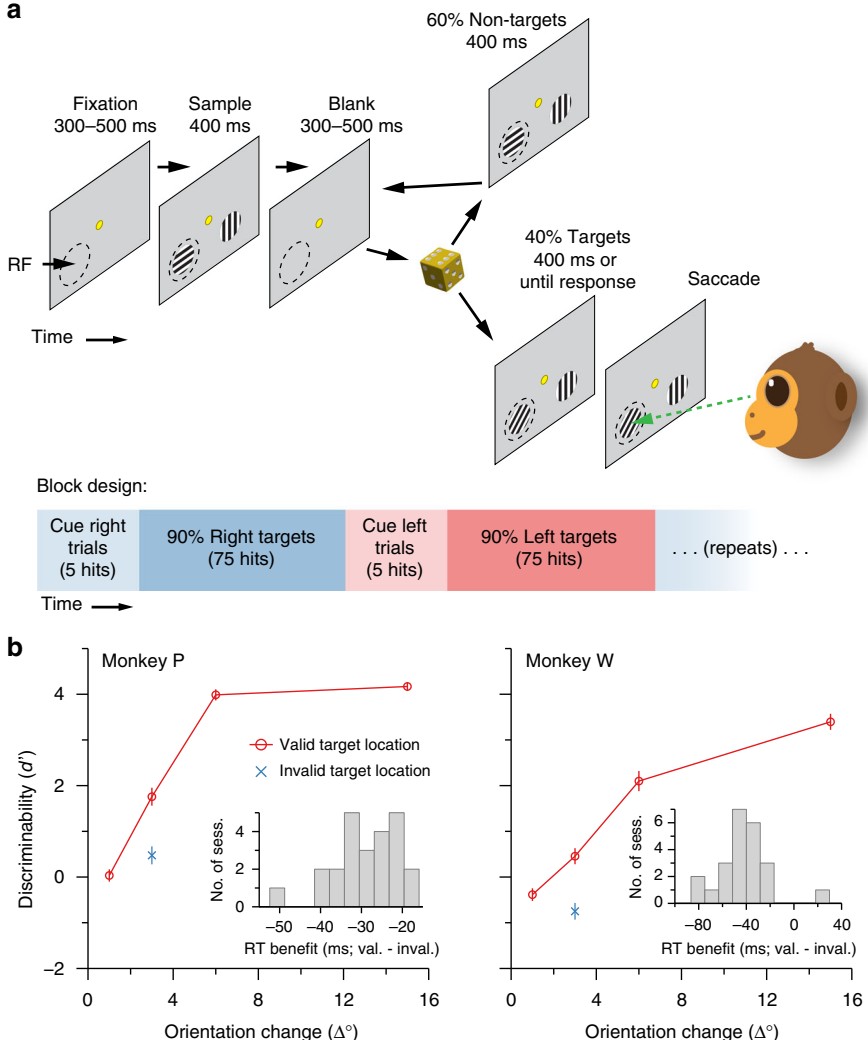

**Fig. 1** Task schematic and behavioral results. **a** Task design. Subjects initiated each trial by fixating their gaze on a central yellow dot. After a 300–500 ms fixation interval, Gabor stimuli were repeatedly flashed for 400 ms, separated by 300–500 ms inter-stimulus intervals. The subjects' task was to detect a change in orientation of one of the two stimuli (the target) and make a saccade to the stimulus that changed. Each stimulus flash had a fixed chance of containing a target (30% for monkey P, 40% for monkey W). One location had a 90% chance of containing the eventual target (the valid location), and we alternated the side of hightarget probability after 80 correct detections (hits). For the initial trials in each block only one stimulus was presented at the valid location (cue trials) until the subject made 5 correct detections, after which bilateral stimuli were presented for the remainder of the block. The dashed circle represents the receptive field (RF) and was not actually present in the display. **b** Behavioral results. For each magnitude of orientation change, we calculated the discriminability index (*d'*). Discriminability was better when targets occurred at the valid location (red circles) compared to when they occurred at the invalid location (blue crosses). Moreover, average response times (insets) were faster for targets at the valid location compared to targets at the invalid location. This pattern of results indicates that the subjects selectively attended to the valid target location at the expense of the invalid location. Error bars indicate ±1 SEM (N = 24 sessions for monkey P and 23 sessions for monkey W)

If the latency of the population average response is too late to underlie the behavioral improvements attributed to attention, then we reasoned that a neural signature of attention may be present in the particular pattern of modulations across the population at behaviorally relevant time points. In other words, the anticipatory effects of attention might not be evident in the population average because at early time points neurons that have consistent decreases of firing rate with attention offset neurons that have consistent increases. Importantly, this means that individual neurons would flip their effects—that is, be suppressed by attention in baseline (leading to minimal population average effects) and enhanced after the stimulus (leading to the canonical gain-like effects). If it was indeed the case that individual neurons changed the direction of their attention modulation between unstimulated and stimulated states, this would present a

particular difficulty for stimulation-invariant models of anticipatory attention, since an invariant gain mechanism would not support suppression at one moment to switch to facilitation later. Even incorporating a divisive normalization mechanism so that the relative magnitude of gain factors could vary across neurons in a population depending on the total amount of activity would not account for a change of the direction of the attention effect, since normalization simply re-scales the effect sizes but does not change their sign. Thus, we next asked whether there were clear examples of neurons suppressed by attention during the pre-stimulus period that were enhanced by attention during stimulus processing.

**Comparing attention modulations with and without a stimulus.** Since we found that simply averaging across the population

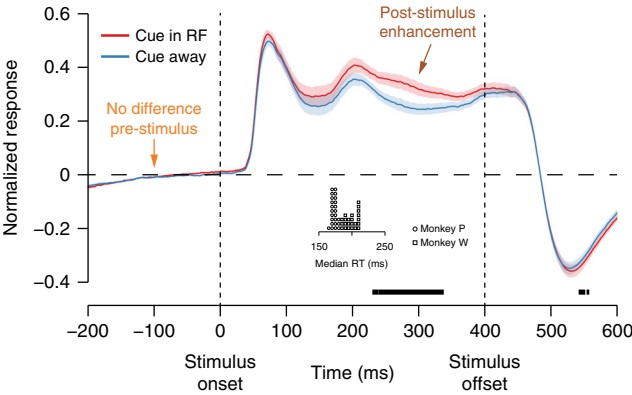

**Fig. 2** Grand-averaged peristimulus spike time histograms (PSTHs). We normalized the responses of each V4 neuron and then averaged over the visually responsive neurons for each session. Responses to sample (non-target) stimuli were stronger when the attention cue was in the RF (red) compared to when the cue was in the opposite hemifield (blue). Shading represents ±1 SEM ($N = 47$ sessions). Black underlining represents a significant difference between attention conditions (repeated-measures *t*-test with α = 0.05, uncorrected for multiple comparisons). The earliest detectable attention effect was not until 233 ms, which was later than the median reaction times for the task (inset pictogram; each symbol represents the median RT for one session; circles for monkey P, squares for monkey W. Note that the time axes for the behavioral and neural data are aligned for comparison)

was inadequate to distinguish attention states at early time points, we next took a closer look at the effects of attention on individual neurons. We found many examples of neurons that were clearly suppressed by attention pre-stimulus, but facilitated during the stimulus response (Fig. 3; see Supplementary Figure 1 for the grand-averaged peristimulus spike time histogram (PSTH) for the subset of neurons with this pattern of effects). Because the direction of the effect changed for these neurons between the two time periods of interest, these observations are difficult to reconcile with a stimulation-invariant gain mechanism even by incorporating normalization in the model. This suggests that there is a meaningful and consistent qualitative shift in the representation of the attention state over time from the pre-stimulus anticipatory period to the stimulus response.

When we examined the distribution of attention effects during the pre-stimulus period over our entire sample, we saw an even split of attentional enhancement and suppression (Fig. 4a, b). Because of this balance, no net effect of attention was evident in the population average response during this anticipatory period (Fig. 2). In contrast, the distribution of attention effects during stimulus processing favored enhancement (Fig. 4b, c), thereby leading to clear enhancement effects in the population average (Fig. 2), consistent with the previous literature. Specifically, the mean increase in sustained, stimulus-evoked firing rate was 9.2% (Fig. 4c), comparable to the findings of Cohen and Maunsell[23], who found gains of 8–10% using a very similar task. Of the 499 neurons with significant attention effects during both time periods of interest, 95 (19%) of them changed the direction of their attention modulation between the unstimulated and stimulated states. It was not the case that attention effects were completely unrelated between pre-stimulus and post-stimulus time periods; however, there was a positive correlation between attention effects measured during the two time periods of interest across the population (Fig. 4b; Pearson's $r = 0.46$, $n = 2591$ units, $p < 0.001$). This observation suggests that at least some of the pre-stimulus attention state is consistent with a gain-like shift at the level of the population, in line with prior reports[13–17], but our

emphasis here was to ask what additional insights might be gained by considering how population patterns of attentional modulation were dissimilar between the stimulated and unstimulated states.

To summarize, we found many individual neurons that showed classical attention-related enhancement of visual responses yet were suppressed by attention in the absence of a stimulus. The mere existence of these neurons indicates that a stimulation-invariant gain mechanism is inadequate to fully characterize anticipatory attention. In order to test the degree to which these novel neural findings have behavioral consequence, we next isolated those aspects of pre-stimulus attention effects that expressly differed from what was seen during stimulus processing, and tested how those novel patterns of modulation related to behavioral performance. This enabled us to assess whether the features of anticipatory attention states that eluded a fixed-gain explanation were behaviorally relevant.

**Pre-stimulus population activity predicts task performance.** To measure the relationships between population activity patterns and behavior, we identified attention axes to quantify the attention state of the population on a moment-to-moment basis (Fig. 5). This approach was previously developed by Cohen and Maunsell[33] to link the strength of attentional modulation on sensory responses in visual cortex to the probability of detecting a subsequent target. Our goal was to test the degree to which unique pre-stimulus activity patterns predict behavior beyond what was already shown possible using stimulus-evoked activity.

An attention axis is a particular vector in the population activity space, which is an *n*-dimensional space where each coordinate axis represents the firing rate of one neuron (Fig. 5 provides an example for $n = 2$). Every point in this space corresponds to a particular set of firing rates across the population. The line connecting the two points in this space corresponding to the trial-averaged set of firing rates for the population observed during the two attention conditions in our task is an attention axis[33,34]. The projection of the population activity onto the attention axis at any given time provides a scalar value reflecting how similar that population activity is to the trial average when the cue was in the RF location or when the cue was away from the RF. These single-trial estimates of attention state can then be used to predict trial-to-trial variability in behavior. For example, if the attention axis projection is meaningfully related to behavior, then the value of the projection before missed targets (which were not used to define the axis) should be reliably shifted away from the value of the projection before correctly detected targets. In contrast, if the attention axis was not meaningful for behavior then the projections would not differ between hits and misses.

We first found the post-stimulus attention axis using the pattern of population activity during the sustained response to the stimulus from 200 to 400 ms following stimulus onset, and then defined a separate, pre-stimulus attention axis using the pattern of population activity during the 200 ms immediately preceding target onset (Fig. 5a). By "separate", we mean that the pre-stimulus attention axis was constrained to be orthogonal to the post-stimulus attention axis (which was held fixed). This is a crucial constraint for the interpretability of our results. By requiring the second attention axis found to be orthogonal to the first, we explicitly ruled out the possibility that similarity of the projections on the two attention axes is due merely to their overlap in the population activity space. Rather, similar projections on the two orthogonal attention axes would only result if more than one direction of variation in the population activity space was actually meaningful for behavior, which would be

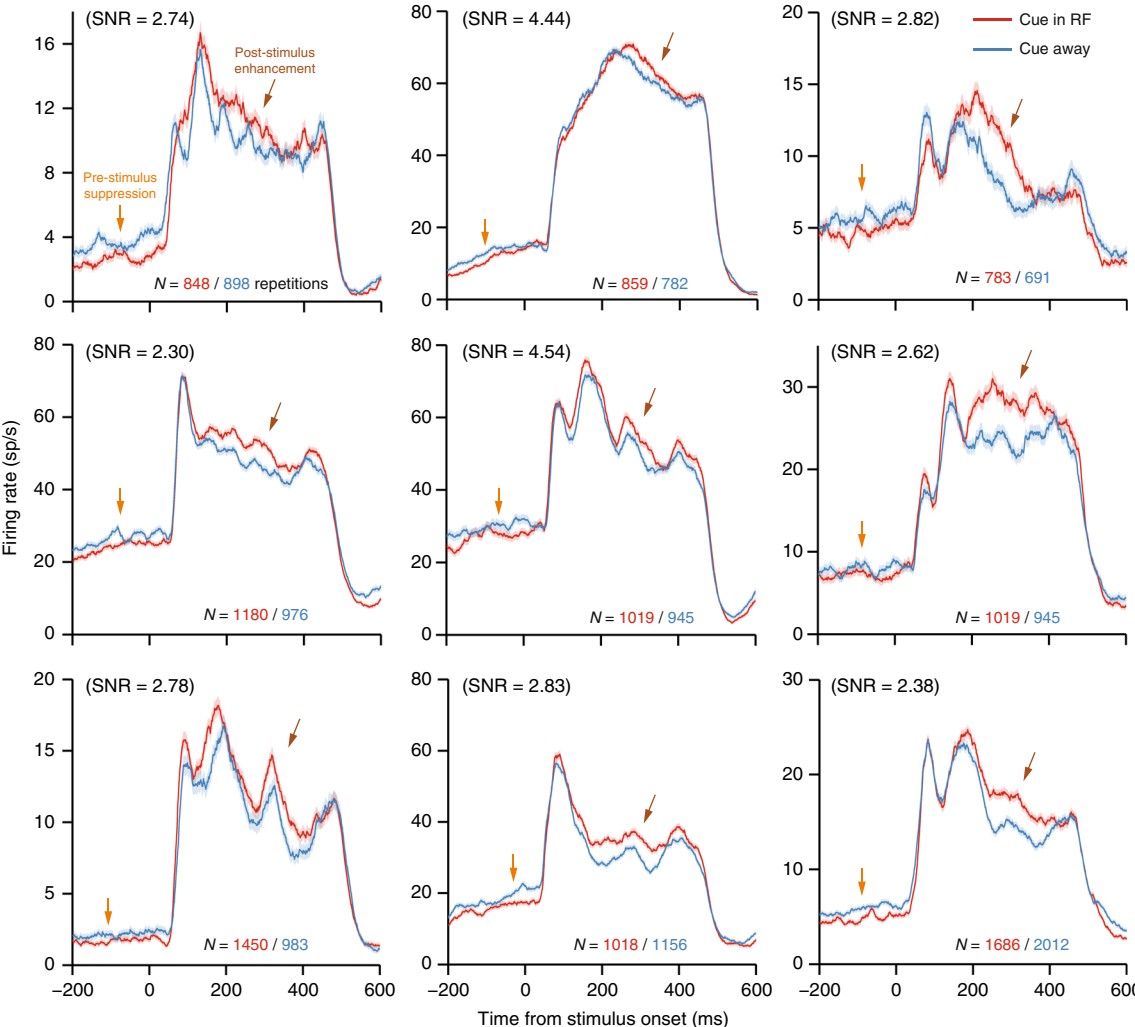

**Fig. 3** Example single-unit PSTHs with dynamic attention effects. All of these single units have significantly greater firing rates during the sustained response to the stimulus (200–400 ms; brown arrows) when the attention cue was in the RF (red) compared to when the cue was in the opposite hemifield (blue), and yet they also have significantly lesser firing rates prior to stimulus onset (−200 to 0 ms; orange arrows) when the cue was in the RF compared to when the cue was out of the RF (two-sample *t*-tests, $\alpha = 0.05$). This pattern of results defies models of anticipatory attention based on a stimulation-invariant gain modulation, but could account for why population average pre-stimulus attentional modulations are typically much weaker than those observed during stimulus responses. For each neuron, the signal-to-noise ratio (SNR) of the action potential waveform is noted. Shading represents ±1 SEM

inconsistent with a stimulation-invariant gain mechanism. The orthogonalization step thus demonstrates that the population pattern of anticipatory attentional modulation is different from the pattern of attentional modulation during stimulus processing, but should not be taken to suggest that the two patterns are precisely orthogonal.

To determine how population activity related to behavior, we examined the neural population activity during the response to the sample stimuli that immediately preceded targets, as well as during the inter-stimulus interval that immediately preceded targets (Fig. 6a). The essential prediction is that on trials for which the subject ultimately correctly detected a target in the RF, the projection of population activity in response to that stimulus onto the post-stimulus attention axis should have been near +1 (indicating attention to the RF by our convention), and when the subject ultimately missed a target in the RF, the attention axis projection prior to the target onset should have been closer to −1 (indicating attention to the non-RF location; Fig. 6b). We also predicted that the same pattern of results would be seen when spontaneous population activity preceding target onset was

projected on the orthogonal pre-stimulus attention axis (Fig. 6c). Because we used cross-validation when determining attention axis projections (see Methods), these results were not guaranteed. We indeed found that the average projections on both attention axes were relatively shifted towards +1 when the RF target was detected, and were relatively shifted towards −1 when the RF target was missed (Fig. 6d). The corresponding pattern was seen for targets out of the RF with the signs reversed, albeit less consistently (Fig. 6e). This latter result is surprising, because it indicates that the V4 neurons accessed by our array in only one hemisphere carried information about the attention state with respect to ipsilateral locations. This could be consistent with a competition for processing resources between the hemispheres (i.e., attention to the RF implies withdrawal of resources for ipsilateral space[35]). Alternatively, our procedure to find the attention axis for targets in ipsilateral space may tap into non-spatially specific attentional processes, such as arousal. The difference in the average attention axis projection preceding detected versus missed targets in the RF was statistically significant for both subjects for both time periods (one-tailed

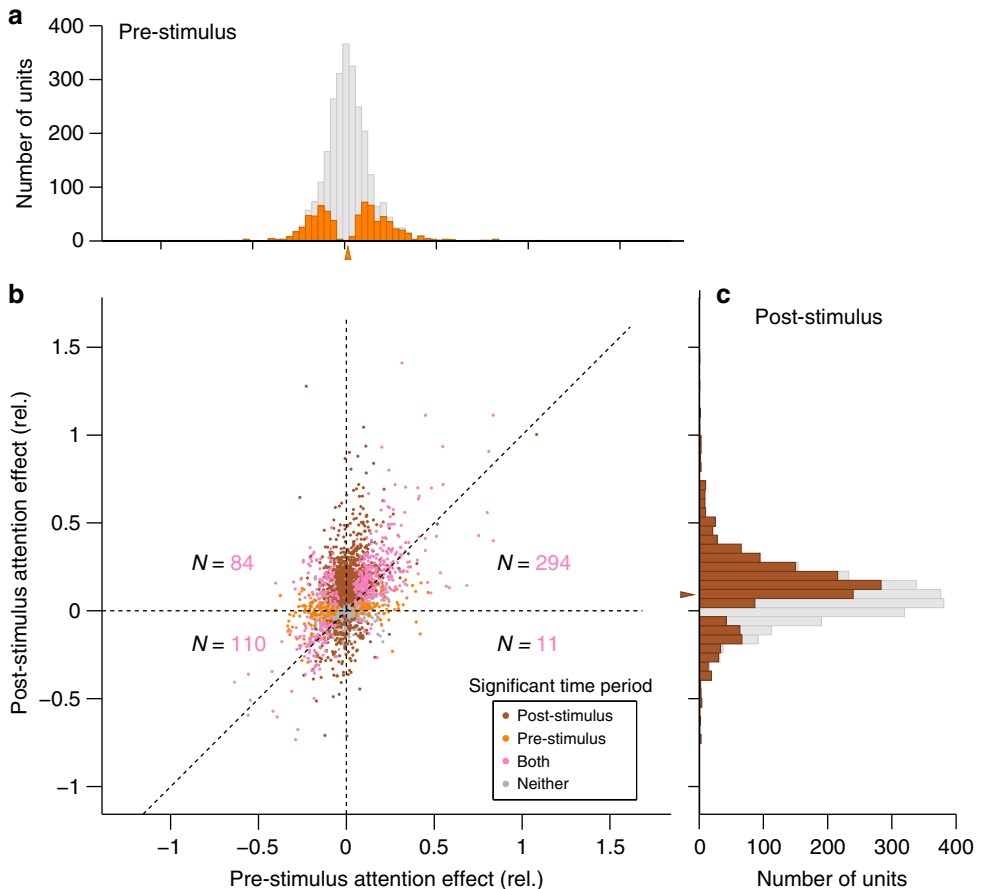

**Fig. 4** Distributions of observed attention effects during both time periods of interest. The relative attention effect is quantified as: ([cue in RF]–[cue away])/[cue away]. Statistical significance was separately assessed during each time interval of interest using Mann–Whitney $U$ tests with $\alpha = 0.05$, uncorrected for multiple comparisons. **a** Distribution of pre-stimulus attention effects. The distribution had near-zero mean, yet many individual neurons were individually significant (orange). Arrowhead indicates mean attention effect value of 0.019 from the full distribution. **b** Joint distribution of attention effects. Note that many neurons had significant attention effects during both time periods of interest (pink), including neurons that were suppressed by attention pre-stimulus yet facilitated by attention post-stimulus (upper-left quadrant). The number of units with significant effects during both time periods in each quadrant is indicated. **c** Distribution of post-stimulus effects. Post stimulus, the distribution of attention effects was clearly shifted towards positive values (brown bars show individually significant units), indicating attention-related facilitation, although a minority of cells did show significant attention-related suppression. Arrowhead indicates mean attention effect value of 0.092 from the full distribution

repeated-measures $t$-tests: pre-stimulus, Monkey P: $N = 24$ sessions, $p = 0.002$, Monkey W: $N = 23$ sessions, $p = 0.002$; post-stimulus, Monkey P: $p < 0.001$, Monkey W: $p < 0.001$). The trend was similar for targets out of the RF, albeit not significant for Monkey P during the pre-stimulus interval (pre-stimulus, Monkey P: $p = 0.083$ not significant (n.s.), Monkey W: $p = 0.039$; post-stimulus, Monkey P: $p = 0.007$, Monkey W: $p = 0.014$). These results show that, at a coarse scale, the attention axis positions that we observed were related to task performance.

The previous analysis revealed that the average attention axis projections differed within a session depending on whether the subject detected or missed the target. We next asked whether attention axis projections explained variation in detection performance from moment-to-moment within each session. For this, we partitioned the full set of trials from each session into 25 subsets of trials based on the projection of population activity during each time period of interest onto the corresponding attention axis (i.e., pre-stimulus activity onto the pre-stimulus attention axis and post-stimulus activity onto the post-stimulus attention axis) and calculated the relative hit rate within each subset of trials (Fig. 7). We found that detection performance was best when the projection onto both attention axes was consistent with attention towards the target location and was worst when the

projection onto both attention axes was consistent with attention away from the target location (Fig. 7a, b). The pattern of results was similar when we used the discriminability index ($d'$) as a behavioral metric instead of hit rate (Supplementary Figure 2). Because we constrained the pre-stimulus attention axis to be orthogonal to the post-stimulus attention axis, this result is inconsistent with a stimulation-invariant gain mechanism. To further illustrate this, we applied our analysis to a simulated dataset in which attention effects were modeled as a stimulation-invariant gain and showed that our analysis does not find a spurious result if orthogonalization is performed (Supplementary Figure 3). Importantly, when we averaged over post-stimulus attention axis projections from the real data to control for that factor, we found that the pre-stimulus attention axis projections continued to explain detection performance for targets in the RF (Fig. 7c, orange). We measured the Spearman's rank correlation between attention axis quintile and hit rate for each session, and found that this relationship was significant for each subject for targets in the RF for both the post-stimulus (Monkey P: mean $\rho = 0.34$, one-sample two-tailed $t$-test: $t_{23} = 5.95$, $p < 0.001$; Monkey W: mean $\rho = 0.20$, $t_{22} = 4.27$, $p < 0.001$) and pre-stimulus (Monkey P: mean $\rho = 0.24$, $t_{23} = 3.92$, $p < 0.001$; Monkey W: mean $\rho = 0.15$, $t_{22} = 2.78$, $p = 0.011$) attention axes,

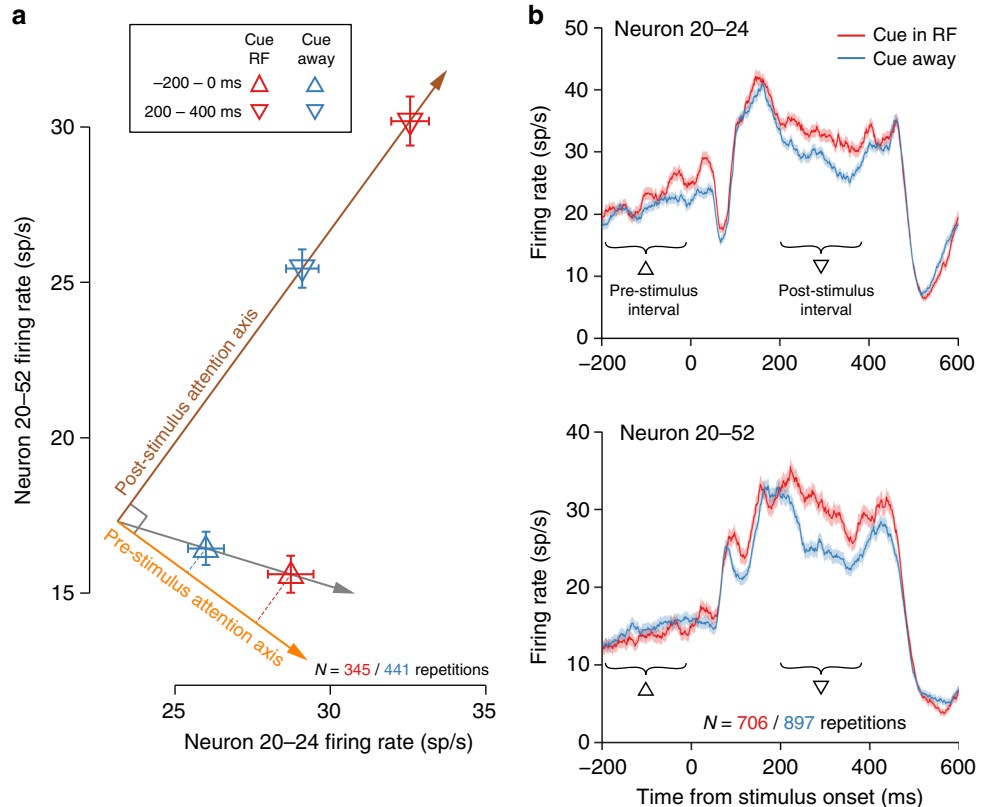

**Fig. 5** Illustration of attention axis identification procedure. In this example only two neurons from our sample are used for illustrative simplicity. In actuality, we used 76.6 ± 11.4 (mean ± SD) neurons per session for monkey P and 32.7 ± 17.8 (mean ± SD) neurons per session for monkey W. **a** First, we identified the post-stimulus attention axis (brown arrow). Each downward-pointing triangle represents the average joint firing rate for the two neurons in response to stimuli that immediately preceded a correctly detected target at the valid location when the RF was the valid target location (red) or when the RF was the invalid target location (blue). The post-stimulus attention axis is the vector through these two points (brown arrow). Second, we found the vector through the average firing rates during the pre-stimulus interval immediately preceding correctly detected targets at the valid location for the two attention conditions (gray arrow). We defined the pre-stimulus attention axis (orange arrow) as the component of the best pre-stimulus vector (gray arrow) that was orthogonal to the post-stimulus attention axis (brown arrow). Note that in two dimensions, as shown, there is trivially only one pre-stimulus attention axis that is orthogonal to the post-stimulus attention axis, but in higher dimensions this is not trivial. Constraining the pre-stimulus attention axis to be orthogonal to the post-stimulus attention axis explicitly eschews a possible interpretation of anticipatory attention based on time-invariant gain (Supplementary Figure 3). That is, we were interested in the degree to which attentional preparation could be captured by a pattern of activity that was linearly independent from that observed in response to a stimulus. Error bars represent ±1 SEM. **b** Peristimulus spike time histograms (PSTHs) for the neurons depicted in (**a**). We averaged the responses to all non-target stimuli on trials that resulted with a correct detection of a target at the valid location for each attention condition. Triangles indicate the time periods of interest as in (**a**). Shading represents ±1 SEM

indicating that each attention axis conveys information about subsequent behavioral performance. Moreover, for RF targets the strength of the relationship between attention axis quintile and detection performance did not differ between the two time periods (repeated-measures $t$-test: Monkey P: $t_{23} = -1.41$, $p = 0.171$; Monkey W: $t_{22} = -0.73$, $p = 0.47$), suggesting that both attention axes have comparable predictive power about the subjects' detection performance for RF targets. In contrast, for targets out of the RF (Fig. 7d), we did not find a consistent significant relationship between the projection on either attention axis and target detection performance (post-stimulus: Monkey P: $p = 0.004$, Monkey W: $p = 0.171$; pre-stimulus: Monkey P: $p = 0.230$; Monkey W: $p = 0.961$). Taken together, these results show that attention states estimated in the presence and absence of a stimulus use distinct patterns of population activity to convey comparable predictive information about the subjects' detection performance for targets at the RF on a moment-to-moment basis.

**Pre-stimulus activity predicts early visual responses.** We set out to determine how attention shapes neural activity to prepare for an upcoming stimulus, having observed that the relatively slow

time course of modulation on evoked responses seems too late to fully account for behavioral benefits (Fig. 2). Reasoning that a key contributor to task performance is the state of readiness preceding the appearance of an anticipated stimulus, we scrutinized the population activity during the pre-stimulus period. We found that a substantial portion of anticipatory attentional modulations were captured by population activity patterns orthogonal to the gain-like modulations seen during stimulus processing, and that these changes added information about subsequent behavioral performance (Figs. 3–7). However, anticipation, while an important antecedent to attentive perception, cannot by itself explain the perceptual improvements of attention, since there is not yet any stimulus to perceive. In a sense, if the gain-like modulations seen in population average responses to a stimulus come too late, anticipatory modulations come too early. If attention modulates neural activity at the time most relevant for behavior, it would make sense that key effects would be focused on the earliest spiking response to a stimulus, when visual neurons convey the most information[36,37].

We therefore assessed how different attention signals measured from the population affected the V4 population activity at

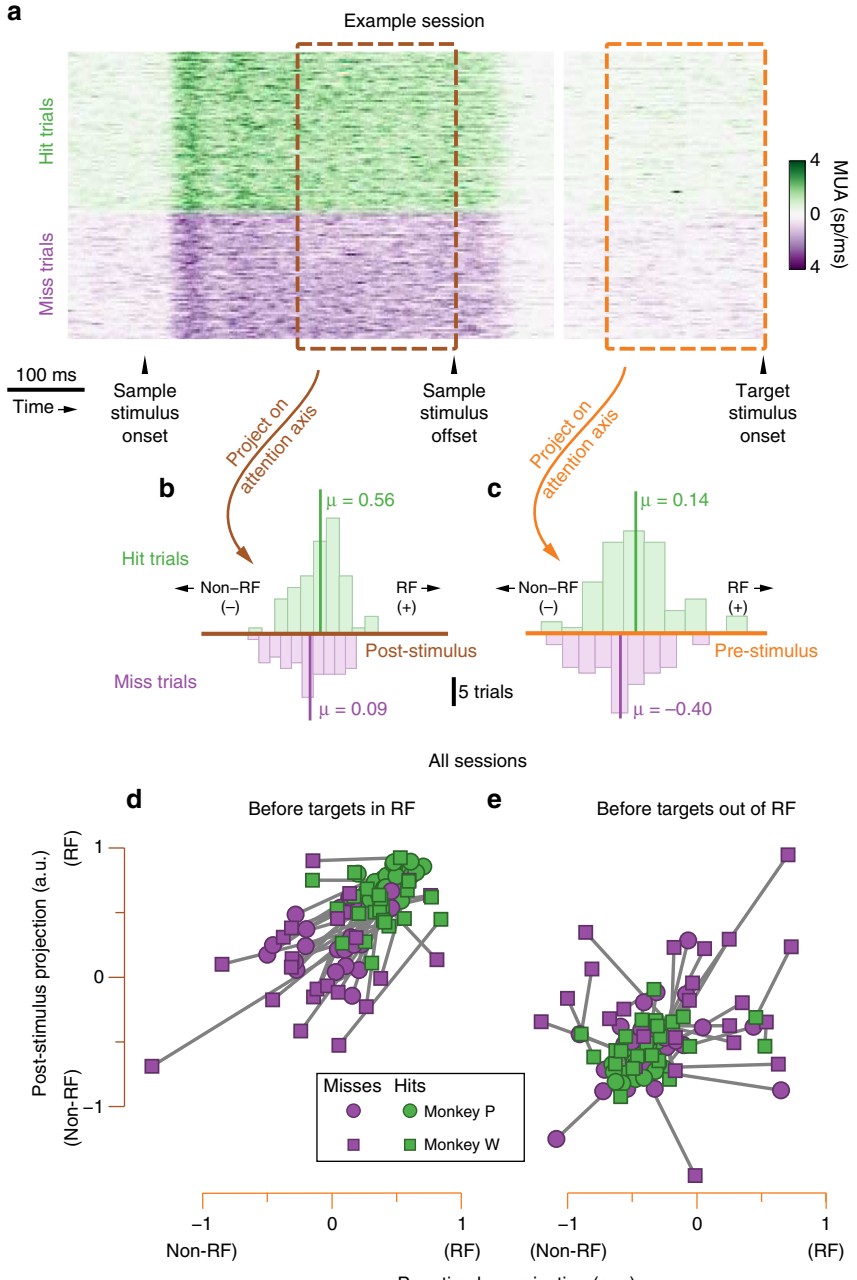

**Fig. 6** Attention axis projections. We separated trials based on the behavioral outcome (hit or miss), and compared distributions of attention axis projections that immediately preceded each target. **a–c** Schematic of analysis strategy with data from example session. **a** We separated trials depending on whether the subject correctly detected the target (green) or missed the target (purple). For illustration only, we combined neurons to show multiunit activity (MUA). **b** We projected the high-dimensional population activity from 200 to 400 ms following the onset of the sample stimulus that preceded each target onto the post-stimulus attention axis (brown) and compared the mean projection when the eventual target was detected (green) to the mean projection when the target was missed (purple; for this example: two-sample one-tailed t-test, $p = 0.011$). **c** We performed the same analysis as in (**b**), projecting the spontaneous population activity during the 200 ms immediately preceding the onset of each target onto the pre-stimulus attention axis (this example: two-sample one-tailed t-test, $p = 0.019$). **d**, **e** Average attention axis projections preceding hit and missed targets for all sessions. Line segments connect observations within each session. **d** When RF targets were detected (green), the average projection that preceded the target was shifted in the positive direction on both attention axes (consistent with attention towards the RF) relative to when RF target was missed (purple). We tested this difference across sessions with one-tailed repeated-measures $t$-tests (pre-stimulus, monkey P: $p = 0.002$; monkey W: $p = 0.002$; post-stimulus, monkey P: $p < 0.001$, monkey W: $p < 0.001$). **e** When targets out of the RF were detected (green), the average projection that preceded the target was shifted in the negative direction on both attention axes (consistent with attention away from the RF) relative to when the non-RF target was missed (purple; one-tailed repeated-measures t-tests: pre-stimulus, monkey P: $p = 0.083$ n.s., monkey W: $p = 0.039$; post-stimulus, monkey P: $p = 0.007$, monkey W: $p = 0.014$). However, compared to targets in the RF (**d**), the average attention axis projections preceding missed targets out of the RF were less consistent

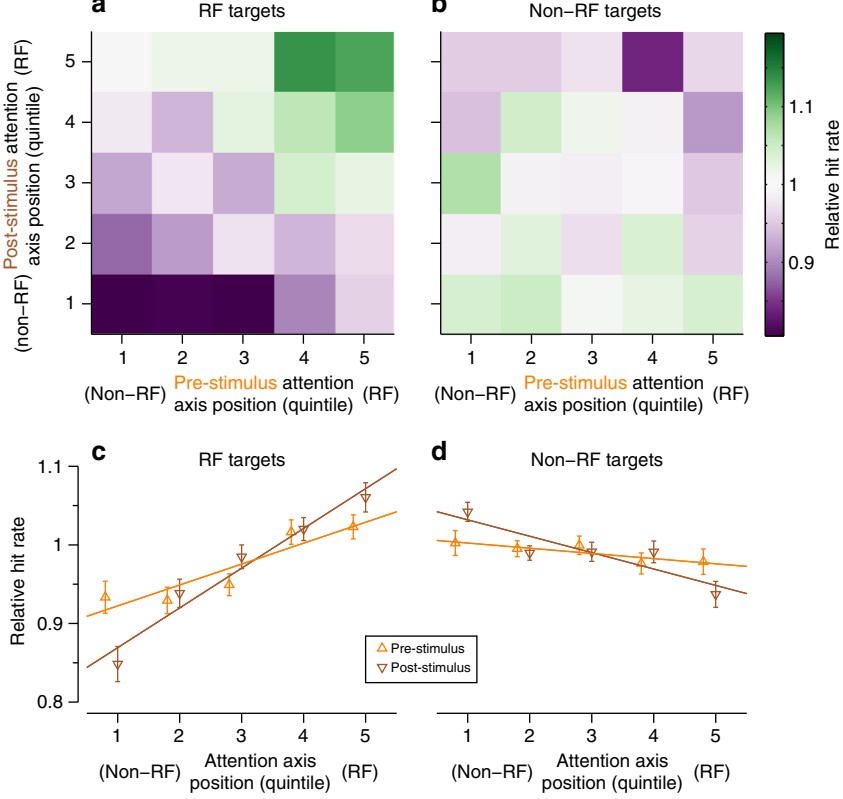

**Fig. 7** Hit rate performance for trials partitioned by attention axis positions. We partitioned the full set of trials from each session based on attention axis positions that preceded targets, and then calculated the hit rate for each subset of trials. **a** Partitioned hit rates for targets in the RF. For each session, we partitioned trials into 25 subsets based on the attention axis projections immediately preceding the target and calculated hit rate within each subset. We normalized the partitioned hit rates within each session by dividing by the overall hit rate for the session, and then averaged over sessions. The best performance (green) was seen when projections on both axes were consistent with attention toward the RF, while the worst performance (purple) was seen when projections on both axes were consistent with attention away from the RF. Within any given bin for one attention axis, variation along the other attention axis was related to performance. **b** Partitioned hit rates for targets out of the RF. The pattern of results was conversely similar to that seen for RF targets (**a**), although the relationship between attention axis positions and performance was weaker. **c** Marginal relationship between pre-stimulus (orange) and post-stimulus (brown) attention axis projections and performance for targets in the RF. Within each session we averaged over the partitions along one attention axis to isolate the unique contribution of the other attention axis. Error bars represent ±1 SEM (N = 47 sessions). **d** As in (**c**), but for targets out of the RF. The relationships were inverted and weaker, relative to those seen for RF targets (**c**)

different times during the response to a stimulus. The key prediction is that if anticipatory attention is important for target detection, then anticipatory attention states should be particularly informative about the earliest part of the sensory response, rather than the later part after a decision has already been made. To test this, we used the attention axis projections described above to estimate the attention state on each trial. Then we asked how much trial-to-trial variance in population stimulus responses were explained by those attention estimates. In particular, we were interested in identifying the time during the response at which any such relationship would be strongest. Because we predicted the population activity pattern and not the overall spike rate averaged across the population, and because we used a continuously variable estimate of attention state rather than a binary division between the two cue conditions, this analysis did not have to show the same time course as the attention modulation of the PSTH (Fig. 2).

We began this analysis by using the post-stimulus attention state to predict the population response to the subsequent stimulus, and we found the strongest predictions around 300 ms after the stimulus onset (Fig. 8, brown). This time course is consistent with the PSTH results (Fig. 2), although we noticed that prediction of the early-latency V4 responses (~50 ms after

stimulus onset; Fig. 8, brown) was improved by these trial-by-trial attention estimates compared to averaging trials over an entire block for each attention condition. Critically, we found we could account for yet more variability in stimulus responses by including the pre-stimulus attention state as a predictor (Fig. 8, orange). Our aim was to compare the timing of when each estimate of attention state provided the most reliable information about subsequent stimulus responses, rather than the absolute strength of that information. In line with our guiding thesis, the additional predictive power of the pre-stimulus attention state peaked early in the visual response, around 71 ms (Fig. 8, orange). This suggests the unique features of population activity patterns during anticipatory attention reflect mechanisms enabling a robust sensory response on the rapid timescales that support brisk and accurate discrimination.

To summarize, we found that pre-stimulus attention states are characterized by patterns of population activity modulations that are fundamentally different from the patterns that characterize attention effects during stimulus processing. Those distinct features of pre-stimulus attention states add unique predictive power about the neural population's response to visual stimulation on behaviorally relevant timescales, and also add unique information about behavioral task performance.

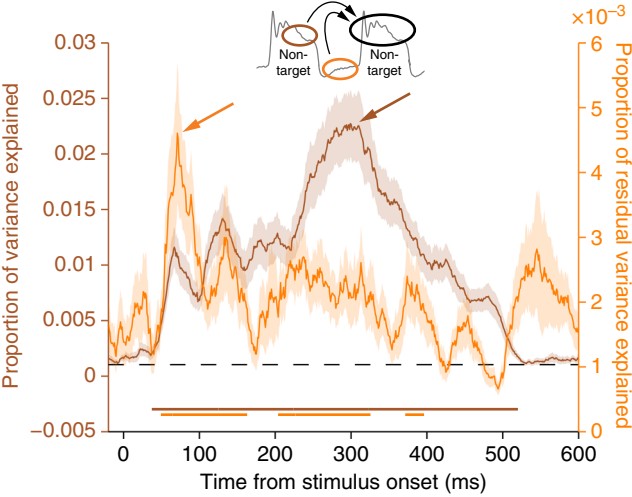

**Fig. 8** The strongest relationship between anticipatory attention states and stimulus responses occurs during the initial transient response. First, we performed a linear regression of the population visual response at each time-point onto the post-stimulus attention axis projection measured during the previous stimulus (brown); the proportion of total variance explained is shown. Second, we performed a linear regression of the residual visual response left unexplained onto the pre-stimulus attention axis projection measured immediately preceding the stimulus (orange); the proportion of residual variance explained is shown. We performed the regression steps in this order to isolate the unique contribution of the pre-stimulus attention state to predictive performance. Both attention axes explain variability in visual responses at earlier time points than was evident when comparing population averages between task block conditions (Fig. 2), although the predictive power of the post-stimulus attention axis peaked around 300 ms (brown arrow). In contrast, the unique contribution of the pre-stimulus attention axis peaked early during the visual response, at 71 ms (orange arrow). The dashed line at VAF = 0.001 represents the average chance value determined by a trial-shuffled control analysis. Shading represents ±1 SEM (N = 47 sessions). Underlining represents a significant difference from chance (one-sample t-test with α = 0.05, corrected for multiple comparisons by requiring 20 contiguous individually significant time points[58])

## Discussion

We found that pre-stimulus anticipatory attention states were captured by patterns of V4 population activity distinct from those that characterized the effects of attention during stimulus processing. This computational framework utilizing different patterns of population activity between stimulated and unstimulated states has advantages over the current prevailing view that anticipatory attention is primarily characterized by baseline gains akin to those seen in response to a stimulus; chiefly, an improved ability for a linear decoder to disentangle internal and external signals from the total population activity. This finding requires reconceptualization of the neural mechanisms underlying the dynamic allocation of attention.

Some previous studies have reported an increase in the average spontaneous firing rate of visual neurons with attention[13–16,18] which we did not replicate. There are many potential explanations for this discrepancy, such as differences in the nature of the task, the nature of the stimulus to be anticipated (which, in our case, could not be optimized to best drive all the cells in our sample), and the criteria for including neurons in the analysis. In particular, we analyzed the attention effects of every cell that we successfully recorded, whereas for other studies it has been common to exclude neurons that are not visually responsive or

not attentionally modulated, and this may have impacted the interpretation of the pre-stimulus attention effects. For one example, Lee et al.[15] restricted their analysis to neurons with at least a 10% attention-related gain in stimulus-evoked responses, and then found that those cells also showed baseline increases. Because there is some direct relationship between pre-stimulus and post-stimulus attention effects (Fig. 4b), conditional selection of neurons based on post-stimulus increases would tend to favor pre-stimulus increases as well. In contrast, our analysis of the entire sample found pre-stimulus attention effects to be roughly evenly divided between increases and decreases.

Although baseline shifts in population average spiking activity are typically slight when they are found, quite large baseline shift effects have been observed in the fMRI blood-oxygen-level dependent (BOLD) signal with attention[26,27]. Even larger anticipatory effects have been seen on the amplitude of alpha-band oscillations of the EEG[28]. Both BOLD signals and EEG signals have been linked more closely to post-synaptic mechanisms than to firing rates[38,39], and hence it is reasonable to suppose that a balanced mixture of anticipatory hyperpolarizations and depolarizations at the level of individual neurons could lead to large BOLD and EEG effects while having relatively little effect on the overall amount of spiking. Combined with the current results, these observations indicate that attention changes the state of neural populations in ways that are (1) metabolically demanding, (2) impact circuit dynamics in the form of oscillations in the field potentials, which likely reflect the input to the sensory area[39], and (3) have relatively little effect on the total amount of spontaneous spiking activity across the population which, in a classically sensory area, could be interpreted as sensory representation. This trio of properties is consistent with our interpretation that attention sets the state of the sensory neural population to be poised to respond, without unduly contaminating the sensory representations of that population.

The idea that top-down attention signals simply impose a stimulation-invariant gain-like amplification on the neural population specialized for processing the attended information is inconsistent with our results. The most extreme violation of this model is that we found individual neurons for which the direction of the attention effect changed from suppressive to facilitatory between anticipatory and stimulus processing time periods. While divisive normalization models can account for a striking diversity of attention effects across a population during stimulus responses[8], this fundamental change between unstimulated and stimulated states cannot be reconciled with such a model. Rather, we propose that top-down attention signals push the population activity into a mode that supports the amplification of sensory representations once a stimulus is presented, but which remains distinct from the patterns of activity that characterize sensory representations until a stimulus appears.

It is well appreciated that the neurons in a cortical area have diverse functional roles. Research classifying neurons based on their primary neurotransmitter and morphological properties has revealed that modulatory influences can depend on the state of ongoing activity[40,41]. Thus, cell class diversity in the population could provide one potential explanation for the diversity of attention effects between the stimulated and unstimulated states. Even within a single class of cells, distinct functional roles for neurons can be conferred based on network topology. For example, neurons within V4 vary in the degree of their coupling to the local population, and neurons with weak population coupling are modulated more by sensory factors than neurons with strong population coupling[42]. We used a simple computational model with two neurons to test whether differential population coupling could provide another potential explanation for the results we observed. One model neuron's firing rate depended

only on stimulus drive and an attention gain factor, reflecting weak population coupling; the other model neuron's firing rate depended only on the rate of the first neuron, reflecting strong population coupling (Supplementary Figure 4). This minimal model was sufficient to replicate the qualitative pattern of results between the stimulated and unstimulated states, and illustrates that differential influence by local or distant inputs could provide a mechanism for diverse manifestations of attention modulation in a population.

Another important source of diversity across neurons is tuning to stimulus properties. Because our task involved discriminating a change in orientation, feature-based attentional strategies could have impacted how the responses of neurons were modulated. For example, neurons preferring orientations at, or near, the orientation of the stimulus at the attended location might exhibit greater enhancement with attention than neurons preferring orientations quite different from that of the attended stimulus[43]. Such interactions between spatial and feature-based attention during stimulus responses have been the subject of substantial previous research[44–49]. Our results indicate that sensory context (i.e., anticipatory vs. stimulus processing states) is another important contributor to diversity in population effects of attention. One drawback of our study design, which used sample stimuli of only two orientations, is that we did not have a compendium of stimuli sufficiently varied to assay how anticipatory attention states interact with stimulus features. Moreover, because the direction and magnitude of the orientation change was unpredictable for our task, a feature-based attentional strategy may have been less fruitful than if a predictable orientation change was used. A future study utilizing a richer stimulus set could help to address this important question regarding anticipatory attention states in neural populations.

Our finding that anticipatory attention utilizes population activity patterns distinct from those that best characterize attention during stimulus processing is reminiscent of recent work in the motor domain, where motor preparation has been shown to use patterns of population activity that are strikingly different from the patterns of population activity observed during motor execution[50–52]. Such parallelism between motor preparation and sensory anticipation suggests the intriguing possibility that the use of different population activity patterns for maintaining separability between internal, modulatory signals in a brain area and the functional output representations of that area (i.e., motor commands or sensory representations) is a conserved computational strategy across the cerebral cortex.

Importantly, we saw that our estimates of the anticipatory attention state accounted for variation in task performance, and that the strength of that relationship was similar to that seen between post-stimulus attention effects and behavior (Fig. 7c). This observation is also inconsistent with a unitary attention mechanism that spans across pre-stimulus and post-stimulus periods. Our observations are consistent, however, with the recent report from Sani et al.[29], who found that pre-stimulus baseline increases were not strongly predictive of post-stimulus attentional modulations of stimulus responses, suggesting that pre-stimulus and post-stimulus effects may be separate manifestations of a latent attentional state yet may not be mechanistically linked.

One question that remains following our study concerns the important transition between anticipatory states and the earliest moments of the sensory response that are essential to rapid and accurate target detection. We supposed that the strength of the population response during the initial onset transient was an important component of this process, and found that anticipatory states were related to the strength of the population onset transient response to non-target stimuli (Fig. 8). However, to directly determine which aspects of the population sensory response link

anticipation to target detection, it would be necessary to analyze the sensory responses to target stimuli. Because the total number of stimuli that can be presented in an experimental session is limited, there is an inherent tradeoff in the design of attention experiments between the number of non-target and target stimuli shown. In this study, we favored inclusion of non-target stimuli, at the expense of fewer targets. The advantage was that we had many presentations of physically identical stimuli uncontaminated by motor responses, but the drawback was that we had too few presentations of any given target stimulus to directly examine which features of population responses were essential for target detection. Future work utilizing a different proportion of target stimuli would help to answer this important question.

Overall, our findings, which incorporate the heterogeneity of neuronal responses across the population and over time within a trial, support a move from considering attention as operating through a simple, stimulation-invariant gain-based mechanism towards a richer view of attentional modulation in a diverse neuronal population.

## Methods

**Ethical oversight.** Experimental procedures were approved by the Institutional Animal Care and Use Committee of the University of Pittsburgh and were performed in accordance with the United States National Research Council's Guide for the Care and Use of Laboratory Animals[53].

**Subjects.** We used two adult male rhesus macaques (*Macaca mulatta*) for this study. Surgeries were performed in aseptic conditions under isoflurane anesthesia. Opiate analgesics were used to minimize pain and discomfort perioperatively. A titanium head post was attached to the skull with titanium screws to immobilize the head during experiments. After each subject was trained to perform the spatial attention task, we implanted a 100-electrode Utah array (Blackrock Microsystems) in V4. We implanted in right V4 for monkey P and in left V4 for monkey W.

**Microelectrode array recordings.** Signals from the arrays were band-pass filtered (0.3–7500 Hz), digitized at 30 kHz, and amplified by a Grapevine system (Ripple). Signals crossing a threshold (periodically adjusted using a multiple of the root-mean-squared noise) were stored for offline analysis. Each waveform segment consisted of 52 samples (1.73 ms). We first performed a semi-supervised sorting procedure to separate putative action potential waveforms from noise. This entailed principal component analysis followed by either an automatic fuzzy c-means clustering algorithm[54] or an automated clustering algorithm based on maximum likelihood estimation of Gaussian mixture distribution parameters (custom functions for MATLAB; The MathWorks, Natick, MA). The automated clustering step was followed by manual refinement using custom MATLAB software[55], taking into account waveform shapes and interspike interval distributions. These initial sorting steps yielded $93.2 \pm 8.9$ (mean ± SD) candidate units per session for monkey P and $61.9 \pm 27.4$ candidate units per session for monkey W. We likely recorded a mixture of single units and multiunit activity, though for simplicity we refer to all units as "neurons". If we quantified recording isolation using signal-to-noise ratios (SNRs) for each unit (defined as the average action potential amplitude divided by the standard deviation of the waveform noise) and restricted our analysis to only units with SNR >3 (to eliminate likely multiunit activity; $29.0 \pm 7.5$ units per session for monkey P, $11.2 \pm 5.6$ units per session for monkey W), the results were not substantively different from the analysis performed using all candidate units (Supplementary Figure 5). Thus, we included all of the single- and multiunit activity from each session (as previous researchers using a similar approach have done[33,34]). To avoid potential confounds due to our blocked design, we excluded neurons that were not recorded stably throughout a session. These were identified by dividing all the recorded data for a session into 10 equally sized blocks, measuring the average firing rate of each neuron within each block, and then calculating the coefficient of variation (CV) of each unit's average firing rate over the blocks. Units with a CV greater than 1 were deemed to be unstable and were excluded. We also excluded neurons with an average firing rate less than 1 spike per second measured over the entire session. These exclusion criteria yielded $76.6 \pm 11.4$ (mean ± SD) units per session from monkey P and $32.7 \pm 17.8$ (mean ± SD) units per session from monkey W. The arrays were chronically implanted and likely recorded some neurons over more than one recording session (although we estimated that a substantial proportion of our sample changed each day; Supplementary Figure 6), but we calculated our results within each recording session and treated each session as an independent sample for the analysis.

**RF mapping.** Prior to beginning the visual change-detection experiment, we mapped the RFs of the spiking neurons recorded on the V4 arrays by presenting small (~1°) sinusoidal gratings at four orientations positioned one at a time in a

grid of positions covering the likely RF area per the anatomical location of the implant. We subsequently used Gabor stimuli scaled and positioned to roughly cover the aggregate RF area. For monkey P this was 7.02° full-width at half-maximum (FWHM) centered 7.02° below and 7.02° to the left of fixation, and for monkey W this was 4.70° FWHM centered 2.35° below and 4.70° to the right of fixation. We next measured tuning curves by presenting gratings at the RF area with four orientations and a variety of spatial and temporal frequencies. For each subject we used full-contrast Gabor stimuli with a temporal and spatial frequency that evoked a robust response from the population overall (i.e., our stimulus was not optimized for any single neuron). For monkey P this was 0.85 cycles/° and 8 cycles/s. For monkey W this was 0.85 cycles/° and 7 cycles/s. For the task, we presented a Gabor stimulus at the estimated RF location, at the mirror-symmetric location in the opposite hemifield, or at both locations simultaneously.

**Visual change-detection task.** Subjects maintained central fixation as sequences of Gabor stimuli were presented in one or both of the visual hemifields, and were rewarded with water or juice for detecting a change in orientation of one of the stimuli in the sequence (the target) and making a saccade to that stimulus (Fig. 1). The probable target location was block-randomized such that 90% of the targets would occur in one hemifield until the subject made 80 correct detections in that block (including cue trials, described below), at which point the probable target location was changed to the opposite hemifield.

The fixation point was a 0.6° yellow dot at the center of a flat-screen cathode ray tube monitor positioned 36 cm from the subjects' eyes. The background of the display was 50% gray. We measured monitor luminance gamma functions by photometer and linearized the relationship between input voltage and output luminance using lookup tables. We tracked the gaze of the subjects using an infrared eye tracking system (EyeLink 1000; SR Research, Ottawa, Ontario). Gaze was monitored online by the experimental control software to ensure fixation within ~1° of the central fixation point throughout each trial. We excluded from analysis data segments during which a subject's gaze left the fixation window.

After fixating for a randomly chosen duration of 300 to 500 ms (uniformly distributed), a visual stimulus was presented for 400 ms, or until the subjects' gaze left the fixation window, whichever came first. For the initial trials within a block, a Gabor stimulus was presented only in the hemifield that was chosen to have a high probability of target occurrence for the block. These cue trials were to alert the subjects to a change in the probable target location and were excluded from the analysis. The initial cue location was counterbalanced across recording sessions. Once a subject correctly detected five orientation changes during the cue trials, bilateral Gabor stimuli were presented for the remainder of the block.

Each trial consisted of a sequence of 400 ms stimulus presentations separated by 300–500 ms inter-stimulus intervals (uniformly distributed). Stimulus sequences continued until the subject made an eye movement (data during saccades were excluded from analysis), or a target was presented but the subject did not respond to it within 700 ms (i.e., a miss). For the first presentation in a sequence, the orientation of the stimulus at the cued location was randomly chosen to be 45 or 135° and the orientation of the stimulus in the opposite hemifield, if present, was orthogonal to this. Subsequent stimulus presentations in the sequence each had a fixed probability (uniform hazard function; Supplementary Figure 7) of containing a target (30% for monkey P, 40% for monkey W), i.e., a change in orientation of one of the Gabor stimuli compared to the preceding stimulus presentations in the trial. Within a block, 90% of targets (randomly chosen) occurred in one hemifield (valid targets) and 10% of targets occurred in the opposite hemifield (invalid targets). For valid targets, the orientation change was randomly chosen to be 1, 3, 6, or 15° in either the clockwise or anti-clockwise direction (monkey P: 11.49 ± 3.14 (mean ± SD, across sessions) valid targets of each orientation at each location; monkey W: 14.56 ± 4.75 valid targets of each orientation at each location). For invalid targets, the orientation change was always the near-threshold value of 3°, clockwise or anti-clockwise (because invalid targets occur infrequently, we restricted the number of orientation change magnitudes for this condition in order to derive a reasonable estimate of the target detection rate). We analyzed trials including either valid or invalid targets, but excluded from analysis all neural data from the time of target onset through the end of the trial.

Monkey R completed 24 sessions of the experiment; monkey P completed 25 sessions. One session for each subject was subsequently excluded from analysis because of recording equipment failure.

**Peristimulus spike time histograms.** From the continuous recording, we extracted data segments from 300 ms prior to stimulus onset to 300 ms following stimulus offset (1 s total segment duration) and counted spikes for each neural unit in 1 ms bins. For the calculation of PSTHs, we smoothed spike trains with a causal half-Gaussian function with σ = 20 ms prior to averaging across trials. To normalize PSTHs for each neuron, we subtracted the baseline firing rate (−100 to 0 ms relative to stimulus onset) and then divided by the absolute value of the most extreme average difference from baseline during the presentation of non-target stimuli, collapsed over all orientations and cue conditions. Because we normalized responses to the maximum deviation from baseline when calculating the population-averaged PSTH, we excluded neural units that did not have a significant visual response (specifically those for which no 50 ms interval during the stimulus had a firing rate different from baseline, as determined by a t-test with α

= 0.05, two-tailed, uncorrected for multiple comparisons). For monkey P, 4.8 ± 1.6 (mean ± SD) units were excluded per session; for monkey W, 1.8 ± 1.6 units were excluded per session. We only excluded these units from the grand-averaged PSTH; visually non-responsive cells were included in the attention axis analyses described below. When testing grand-averaged PSTHs for a difference between attention conditions using t-tests (Fig. 2), we confirmed firing rate distributions did not violate a normality assumption using Lilliefors' test.

**Attention axes.** The concept of an attention axis in the population activity space has previously been used to quantify attention states on a single-trial basis during stimulus processing[33,34]. Here we extended this concept by also identifying a unique attention axis during pre-stimulus, preparatory activity.

The population activity space is an n-dimensional space in which each coordinate axis represents the firing rate of one neuron (Fig. 5 shows an example for n = 2). Each point in the space corresponds to a particular set of firing rates across the neurons in the population. An attention axis is the line connecting the two points in this space that correspond to the trial-averaged set of firing rates observed during each of the two attention conditions in our task (i.e., cue in RF vs. cue away from RF). Once an attention axis is identified, the set of firing rates across the population observed at any given time can be projected onto that axis, resulting in a scalar value that reflects how similar the pattern of population activity is to the trial average during the cue-in-RF condition or the cue-away-from-RF condition.

To define the attention axes, we analyzed population responses immediately preceding validly cued and correctly detected targets (i.e., hits), reasoning that these intervals were most likely to have been attended due to the favorable behavioral outcome. To ensure no data were ever used to simultaneously define an attention axis and determine the position of the population activity along that same axis, we used cross-validation. Specifically, we found the attention axes using half of the trials, selected randomly, and then projected the population activity of the held-out trials onto the axes identified. We iterated this procedure 1000 times, randomly selecting a new held-out trial set each iteration, and then averaged the attention axis projections over cross-validation folds. Because the magnitude of the attention axis projections depends on task-irrelevant factors such as the number of neurons in the sample, when attention axis projections were compared across sessions (i.e., Fig. 6d, e), we normalized the projections of all trials within each cross-validation fold so that the average projection of the trials from the cue-in-RF condition used to define the axes (i.e., excluding the held-out trials) had a value of +1 and the average projection of the trials from the cue-away-from-RF condition used to define the axes had a value of −1. Note that because we used cross-validation, the average attention axis projections for validly cued hit trials reported are not guaranteed to be ±1 (Fig. 6). When performing statistical tests on attention axis projections, we confirmed that distributions did not violate an assumption of normality using Lilliefors' test.

We defined attention axes using population activity during two epochs: (1) a pre-stimulus attention axis defined using the responses during the 200 ms immediately preceding target onset, and (2) a post-stimulus attention axis defined using the responses from 200 to 400 ms following the onset of the stimulus that preceded the target. We always projected data onto the axis defined during the corresponding time period of interest (i.e., pre-stimulus data onto the pre-stimulus attention axis and post-stimulus data onto the post-stimulus attention axis). Critically, our guiding prediction was that the population pattern of attention modulations during anticipation is qualitatively distinct from the pattern seen after stimulus onset. To test this, it is necessary to rule out the potential interpretation that the pattern of firing rate modulations that distinguishes pre-stimulus attention states is just a scaled version of the pattern of firing rate modulations that distinguishes post-stimulus attention states. Thus, after we identified both attention axes (and prior to projecting population firing rates onto those axes) we orthogonalized the pre-stimulus axis with respect to the post-stimulus attention axis using the MATLAB "QR" function on the basis spanned by the two attention axes (Fig. 5; although there is trivially only one axis orthogonal to the post-stimulus attention axis when N = 2, as in Fig. 5, in general there are infinitely many orthogonal axes when N > 2, so finding the best orthogonal pre-stimulus attention axis is not trivial). For reference, the average angle between attention axes prior to orthogonalization was 74.3 ± 15.5° (mean ± SD) for monkey P, and 79.5 ± 17.9° for monkey W. Because the pre-stimulus attention axis was constrained to be orthogonal to the post-stimulus attention axis, it necessarily reflects a distinct population pattern of firing rate modulations from that observed during the stimulus response. To further illustrate this point, we performed our analysis on a surrogate dataset in which attention modulations were generated from a model reflecting a stimulation-invariant gain mechanism, and found that our approach did not lead us to find a spurious pre-stimulus attention effect in that case (Supplementary Figure 3).

**Correlation between attention axis projection and behavior.** To quantify the strength of the relationship between attention axis projections and behavior within a session, we binned the projections on each axis during intervals that preceded targets into quintiles, yielding 25 bins total across both axes, and calculated the hit rate in each bin (Fig. 7). Then, for each session we measured the Spearman's rank correlation between bin index and hit rate for each axis. For statistical analysis of the resulting rank correlation coefficients, we first applied the variance-stabilizing

Fisher's $r$-to-$z$ transformation. We confirmed that the distributions of the $z$-transformed correlation coefficients did not violate an assumption of normality using Lilliefors' test. Then, we tested the distribution of rank correlation coefficients for each subject against a null hypothesis of zero correlation using a one-sample, two-tailed Student's $t$-test with $\alpha = 0.05$. We tested for differences in the strength of the rank correlations between the two attention axes with a repeated-measures, two-tailed Student's $t$-test with $\alpha = 0.05$. After calculating the mean of the $r$-to-$z$-transformed rank correlation coefficients, we applied the inverse $z$-to-$r$ transform on the mean value for reporting purposes.

**Two-step linear regression analysis**. To quantify how well attention axis projections at a given time predicted the strength of the population response to a stimulus at a later time, we performed a two-step ordinary least squares linear regression analysis. We used a two-step approach to conservatively estimate the unique contribution of our novel pre-stimulus attention axis after we had accounted for as much variance as possible using the post-stimulus attention axis projection, which reflected attention effects that were already well established in the literature.

We derived population stimulus responses using the following method. First, we extracted spiking responses from 100 ms prior to the onset of each stimulus until 200 ms after stimulus offset. We smoothed the spike trains for each neuron with a causal half-Gaussian window with a standard deviation of 20 ms. We then summarized the strength of the population response to each stimulus as a one-dimensional time series by projecting the smoothed population activity from each stimulus presentation onto a stimulus response axis in the population activity space. We defined the stimulus response axis as the line in the population activity space that connected the trial-averaged baseline activity (over the $-100$ to 0 ms relative to stimulus onset) to the point where the trial-averaged population activity was furthest from the baseline level (the high-dimensional analog of the peak of the onset transient).

We restricted our analysis to stimuli that satisfied two conditions: (1) they must have been a sample stimulus (not a target), since eye movements to targets would contaminate the sensory response, and (2) they must have been preceded by another sample stimulus on the same trial, since the first stimulus in a trial sequence is preceded by an eye movement to achieve central fixation that could contaminate the neural activity. We define $r_{ti}$ as the one-dimensional population response at time $t$ (1 ms bins) to the $i$th of $n$ stimuli to be predicted, and $a_i^{post}$ and $a_i^{pre}$ are the attention axis projections measured during the stimulus response or inter-stimulus interval, respectively, that *preceded* the $i$th stimulus to be predicted. We performed our analysis time-point by time-point, with $\mathbf{r}_t$ a mean-centered $1 \times n$ vector of population stimulus responses at time $t$, and $\mathbf{a}^{post}$ and $\mathbf{a}^{pre}$ $1 \times n$ vectors of attention axis projections. We first found a scalar regression coefficient at each time-point, $\hat{\beta}_t^{post}$, to predict stimulus responses from post-stimulus attention axis projections:

$$\hat{\beta}_t^{post} = \arg\min_{\beta_t^{post}} \left\| \mathbf{r}_t - \mathbf{a}^{post}\beta_t^{post} \right\|^2 \tag{1}$$

The proportion of variance accounted for (VAF) in the population stimulus response by this model (Fig. 8, brown) is given by:

$$VAF_t^{post} = 1 - \frac{\left\| \mathbf{r}_t - \mathbf{a}^{post}\hat{\beta}_t^{post} \right\|^2}{\mathbf{r}_t^2} \tag{2}$$

For the second step of the regression analysis, we asked whether we could account for residual error from the first regression step, $\eta_t = \mathbf{r}_t - \mathbf{a}^{post}\hat{\beta}_t^{post}$, using the pre-stimulus attention axis projection as a predictor variable:

$$\hat{\beta}_t^{pre} = \arg\min_{\beta_t^{pre}} \left\| \eta_t - \mathbf{a}^{pre}\beta_t^{pre} \right\|^2 \tag{3}$$

The proportion of residual variance in the population stimulus response accounted for by this second model (Fig. 8, orange) is given by:

$$VAF_t^{pre} = 1 - \frac{\left\| \eta_t - \mathbf{a}^{pre}\hat{\beta}_t^{pre} \right\|^2}{\left\| \eta_t \right\|^2} \tag{4}$$

To test for statistical significance of the proportions of VAF, we compared our results to the results of a control analysis in which the order of the attention axis projections was randomly shuffled within each session. If both the predicted and predictor variables of a regression analysis are Gaussian-distributed and statistically independent, the expected proportion of VAF is $1/(n-1)$, where $n$ is the number of observations which would be stimulus presentations in our case[56]. We found that the result of our shuffled analysis was very close to this theoretical value and did not depend on time during the visual response, suggesting that our data were in line with the assumptions of the theoretical null hypothesis value for VAF. We averaged the results of the shuffled control analysis over time and over sessions,

yielding $VAF^{null} = 0.001$. We then tested our observed $VAF^{post}$ and $VAF^{pre}$ at each time-point for a difference from $VAF^{null}$ using a one-sample $t$-test with $\alpha = 0.05$ and $N = 47$ sessions (distributions were normal by Lilliefors' test). We corrected for multiple comparisons by requiring a minimum run length of consecutive significant time points. This correction relies on the fact that increasing numbers of consecutive significant $t$-scores become increasingly unlikely under the null hypothesis. The run length threshold was determined with a simulation using the autocorrelation of the VAF time series (since adjacent samples are not entirely independent) to determine the minimum run length such that the false discovery rate for finding a single effect over all time points under the null hypothesis was the desired 5%[57,58], which in this case we found to be 20 samples.

## Data availability

Analysis computer code and data for this project are available from the authors upon request.

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

## Acknowledgements

A.C.S. was supported by NIH grant K99EY025768. M.A.S. was supported by NIH grants R01EY022928 and P30EY008098, Research to Prevent Blindness, and the Eye and Ear Foundation of Pittsburgh. B.M.Y. was supported by NSF NCS BCS1533672, NIH R01 HD071686, NIH CRCNS R01 NS105318, and Simons Foundation 364994. B.M.Y. and M.A.S. were supported by NSF NCS BCS 1734916/1734901. The authors would like to thank Ms. Samantha Schmitt for assistance with surgery and data collection.

## Author contributions

Conceptualization: A.C.S., M.A.S., and B.M.Y.; methodology: A.C.S. and M.A.S.; software: A.C.S.; formal analysis: A.C.S.; investigation: A.C.S.; resources: M.A.S. and B.M.Y.; writing (original draft): A.C.S., M.A.S., and B.M.Y.; writing (review and editing): A.C.S., M.A.S., and B.M.Y.; visualization: A.C.S.; supervision: M.A.S. and B.M.Y.; project administration: M.A.S. and B.M.Y.; funding acquisition: A.C.S., M.A.S., and B.M.Y.

## Additional information

**Competing interests:** The authors declare no competing interests.

