## [Peer Review File · Nature Communications]

Reviewers' comments:

Reviewer #1 (Remarks to the Author):

This is an interesting paper, which reports that during stimulus expectation top-down spatial attention alters baseline firing rates in different ways across the neuronal population. Specifically, a subset of neurons shows decreased baseline activity, when they represent the attended location.

There are a few main points that I think should be addressed:

1. While the results are interesting, the study does not address how this mixed model could quickly aid the detection of the relevant stimulus. If the attentional modulation is too sluggish to explain reaction times, then how does attention aid behaviour? The argument put forth is that the pattern of modulations across the population might aid the behaviour. But if some neurons switch their sign of attention effects after stimulus presentation, this switch may be at a later time point, hence these might initially result in decreased attention effects across the population, which then averages out to none for the first 150 ms, because of the 'other population'? This is somewhat supported by visual inspection of the example cells provided. Many of these show strong attentional effects (of either sign) in the early phases of stimulus onset. What does population activity look like, when split between (a) neurons showing pre-stimulus attention suppression (and switching), (b) pre-stimulus enhancement, no switching after stimulus, (c) no prestimulus effect, post stimulus attentional effect?

Moreover, modulations may be different in specific ways for the target stimulus, especially in populations best suited to detect subtle orientation changes, ie. those where the steepest part of the tuning curve falls at the sample-target orientation difference. A way to address this might be analysis of responses to the target across all neurons and across those to which the previous applies. While this can be affected by the saccade, it should be possible to analyse at least the first 150 ms.

2. The authors argue that this mixed gain change, can disentangle the effects of attention and the effect of a weak stimulus being presented, by ensuring that some cells take on negative weights during anticipatory attention in downstream areas. I think this needs at least clarification in terms of how the authors think that is implemented, and how a negative weight could be switched to a positive weight after stimulus presentation?

3. I think a real statistical problem is to treat units recorded across session as independent observations. Given the Utah array recordings, this is extremely unlikely to be the case, and thus re-sampling is a real issue, and the implications on statistics. This is most prevalent when the effects are calculated for a session (e.g. attention axis), and then the p-value is based on the number of session. Assume (worst case scenario), the same neurons were recorded every session. Then the same effect would have to occur from session to session, and the p-values would be meaningless. I think it is important to establish, to what extent neurons differ between sessions, or think about different ways to address this problem.

Minor:

4. The authors argue that the unattended hemisphere has knowledge about the attentional state of the attended hemisphere. This is maybe too strong an argument. If attend away leads to a different state than no attention, and yet than attend to, the finding that the neurons can decode between hits and misses in the unattended location, may simply be a result of attentional lapse (no attention). And this would possibly be expected to be less pronounced than differences between attend RF and misses, if attentional 'down modulation' during attend away was less pronounced than up modulation in attend RF.

5. Typo in methods, line 605: If the both the....

Reviewer #2 (Remarks to the Author):

This paper examines the neural correlates of “anticipatory attention” in the dorsolateral prefrontal cortex (dPFC), during an epoch in a change detection task in which the attentional locus is cued but before a stimulus appears. The results show that the locus of attention can be read out from the population response in dPFC using linear decoding, and that this readout is independent of the gain changes that are observed after stimulus presentation.

The paper is potentially interesting, but there are several major deficiencies in the analysis that make the results questionable.

1. The main argument of the paper is that there are population effects in the pre-stimulus response that are orthogonal to those in the post-stimulus response and capture some of the behavioral variance. However, throughout the paper we are never told *how much* variance – neural and behavioral - is captured by these components. Without this information it is very difficult to estimate the significance of the results.

2. This concern is highlighted by the only mention of percent variance explained (PEV) that comes in the last figure (no figure numbers are provided). That figure is inscrutable (very difficult to follow, along the lines described below) and the axes only go up to 3% of total variance explained (up to 0.006 of the residual variance) which seems like a ridiculously small number.

3. The analysis apparently uses multi unit and single unit data, making the conclusions highly questionable, especially given the very small effect sizes.

4. The penultimate figure shows correlation between neural projections and hit rates, but this is confounded with response bias. The appropriate analysis should use d' .

5. The authors use t-tests to look at firing rate differences without even a mention of whether they did normality tests. This use must be justified or replaced with non-parametric tests.

Compounding these concerns, the writing suffers from a pervasive lack of clarity that makes the paper very frustrating to read and understand.

1. The task and analyses are carelessly described. Monkeys are trained on a task in which, in alternating trial blocks, they must attend to a Gabor stimulus in the right or left hemifield and make a saccade when they detect a change in this stimulus. Stimuli flash on and off (400 ms on, 300-400 ms off) and the change has a 40% probability of happening on each cycle after the initial “sample”. The focus of the paper is on activity before stimulus onset – during the “off” period – but throughout the paper there is a lot of confusion about which one of the several off periods, this analysis is applied to. First, it would help to have a histogram of the number of cycles before a response across all the trials in their data set. Second, the authors should be explicit about which one(s) of the off epochs they include. Third, they should discuss – and rule out – any effects of hazard rate – the monkeys’ increasing anticipation of a change as time progresses. The latter point is very important and may change both behavior and neural responses. Fourth, they should discuss whether they include valid and invalid trials since the saccade direction differs in these cases (if both trials are correct).

2. Confusion is particularly high in the figure comparing hit and miss trials (there are no figure numbers). That analysis suggests that the monkey makes similar number of hits and miss trials in a session. How does this reconcile with the high d' values in Fig. 1? Are these trials selected in some way? And what is the “pre-stimulus” epoch and the “pre-target epoch”? How do these relate to the epochs in the previous figures? And which attention axes are used to calculate the projections? Is pre-stimulus activity (or pre-target activity) projected on the post-target axes? (and if so, what is the justification, given that we are interested in the pre- and post-target components that are orthogonal to each other?)

3. Similar concerns – too many to list – come up for the remaining figures.

Reviewer #3 (Remarks to the Author):

Substantial work has demonstrated that selectively attending to one of several objects will increase the firing rate of neurons tuned to the location of the objects within the visual field. In some studies (almost always in fMRI, sometimes in EEG/MEG, rarely in electrophysiology), changes in pre-stimulus baseline activity levels is also observed – typically such that greater activity precedes the presentation of a stimulus in an attended part of space. But, given that these effects are challenging to see in electrophysiological recordings, and are often obscured when averaging across all recorded neurons, perhaps the typical story that attention induces a gain in responses, regardless of stimulation, is incomplete.

The authors found striking evidence for an alternative model that allows for different types of modulation (i.e., suppression vs enhancement) at different phases of the trial, within the same individual neuron. Such results are demonstrated to be unlikely to occur due to a simple stimulation-independent gain model, are related to behavioral performance, and allow for independent readout of pre-stimulus attentional state and post-stimulus attentional enhancement of stimulus representations.

I think this is a really excellent report which the readership of Nature Communications will find fascinating. The question is novel and important, the analyses employed are elegant, the data is presented thoroughly, and the writing is exceptionally clear. In particular I very much appreciate Figure 4, which shows the full set of modulations across the dataset. These figures are somewhat rare in the literature, but are also so important for getting a handle on the true expected distribution of these effects.

It's always a pleasure to review a report that is so carefully and thoughtfully prepared. I have a few conceptual questions, mostly out of my own curiosity, and a couple of very minor suggestions that could improve the presentation of the results. Perhaps the authors may be interested in adjusting the manuscript to include some of the conceptual issues I raise, but I emphasize that this would be purely optional and the choice not to include such text would not in any way impact my judgment of a revision, should that be necessary.

1. The authors estimated RFs/tuning functions for each neuron. The differences in orientation the monkeys were required to detect were quite small – this suggests that the animals could perform better if they preferentially allocated gain to neurons tuned slightly away from the non-target orientation values (see, for one of many examples in the human literature, Scolaro et al, 2012, J Neuro). Without emphasizing this particular possibility too much, I'm curious if the authors have evaluated whether orientation preferences modulate the attention gain effects they see (either pre or post stimulus)? This is already quite a comprehensive manuscript which sufficiently makes its key points, so I leave it to the authors' discretion whether to explore and/or discuss such a possibility.
2. Related – a pure gain increase may not optimally enhance the amount of information about the relevant sensory variable (orientation). Perhaps the authors might care to discuss the relationship between response intensity/spike rate and information content of the population code, and how attention can impact one, the other, or both? Was this issue examined in the present dataset? I imagine the task design makes this somewhat challenging – since the target stimuli necessarily result in a very fast oculomotor response, all analyses must be focused on the samples. In any case, I'm curious what the authors think about this. A potentially relevant report in humans is Myers et al, 2015, eLife.
3. Finally, one last conceptual issue related to orientation preference: does the pre-stimulus change in

firing rate depend on orientation tuning and the 'sample' orientation within the RF on that trial? If so, this could speak to a particularly strategic re-apportioning of metabolic resources by attention on a trial-by-trial basis.

4. In Figure 4 – do the brown/orange arrowheads indicate the mean of the full distribution, or the mean of the significant neurons within each marginal distribution?

5. Figure 6 – the small insets in panels b & c – what is the x range on each of those? -1 to +1?

6. I'm curious how orthogonal the pre- and post-stimulus attention axes are pre-orthogonalization? (what is their vector angle?).

7. Figure 5a – I'm not sure the 2-d case is the most optimal to show since there is only a single possible orthogonalized pre-stimulus attention axis for any 2d post-stimulus attention axis. Would a 3d example be possible? In this case (if I'm understanding correctly), the pre-stimulus attention axis would still lie along the plane defined by the post-stimulus attention axis and the 'raw' pre-stimulus attention axis (gray arrow), but will be oriented orthogonal within that plane. That is – the gray arrow restricts which of the infinite possible orthogonal vectors is called the pre-stimulus attention axis. (and I do apologize if I'm a bit lost confused here – I'm happy to be corrected by the authors)

8. The authors may wish to add a code/data availability statement. I would encourage they publicly deposit data/code in an easily-accessible permanent repository like Open Science Framework (osf.io)

We are grateful for the comments from the three reviewers of our manuscript, and for the overall enthusiasm for the importance of the ideas we present. The reviewers' questions focused on a few main issues: (1) More information about the mixture of individual neuronal signals that constituted the diverse attention signal we observe, (2) Analysis of the orientation-specificity of attention effects (and thereby the potential impact on the behavior in the task) and (3) Better clarity in our presentation of some of the technical details of our analyses. To address these points, we have added 5 new supplemental figures and made substantial textual edits throughout the manuscript (changes marked in red in the manuscript). We believe that the revised manuscript is stronger and clearer as a result. Specific replies to each reviewer comment are shown below in blue text.

Reviewers' comments:

Reviewer #1 (Remarks to the Author):

This is an interesting paper, which reports that during stimulus expectation top-down spatial attention alters baseline firing rates in different ways across the neuronal population. Specifically, a subset of neurons shows decreased baseline activity, when they represent the attended location.

There are a few main points that I think should be addressed:

1. While the results are interesting, the study does not address how this mixed model could quickly aid the detection of the relevant stimulus. If the attentional modulation is too sluggish to explain reaction times, then how does attention aid behaviour? The argument put forth is that the pattern of modulations across the population might aid the behaviour. But if some neurons switch their sign of attention effects after stimulus presentation, this switch may be at a later time point, hence these might initially result in decreased attention effects across the population, which then averages out to none for the first 150 ms, because of the 'other population? This is somewhat supported by visual inspection of the example cells provided. Many of these show strong attentional effects (of either sign) in the early phases of stimulus onset. What does population activity look like, when split between (a) neurons showing pre-stimulus attention suppression (and switching), (b) pre-stimulus enhancement, no switching after stimulus, (c) no prestimulus effect, post stimulus attentional effect?

The question of what happens during the onset transient to aid target detection is indeed a critical one. We tried to address this question indirectly by looking at how anticipatory attention states affect responses to non-targets (Figure 8). We found that the attention state of the population, as indexed by the projection of activity along either attention axis, was predictive of the strength of the population response to a subsequently presented stimulus. The interesting result is that the pre-stimulus attention axis was particularly informative of the strength of a subsequent onset transient (rather than the later, sustained response), which we reasoned was the behaviorally relevant period of the response. Thus, our results suggest that anticipatory attention places the population in a state conducive to a strong onset transient response to a subsequent stimulus, which may lead to better target detection. However, because this analysis was performed on non-target stimuli (which were plentiful and uncontaminated by motor responses in our task), the link to target detection is through inference only. In designing our study we were faced with a trade-off between the number of targets vs. non-target stimuli that we presented, an inherent trade-off in any change-detection task. We chose to favor non-targets with the advantage that they were all physically identical, which provided many repeated presentations even after partitioning trials into subsets based on attention axis projections. If we increased the proportion of targets presented, this would have decreased our power to analyze responses to non-targets.

However, the improvement in power for analyzing responses to targets would not have equally offset that cost, since it would have to have been split across the eight different orientation change magnitudes that we used. Thus, we opted to favor the presentation of non-target stimuli for this study. We concur with the reviewer that a detailed understanding of the change in target-processing that underlies behavioral improvements is of great importance, and would be a fitting focus of future inquiry. We now discuss this issue in the revised Discussion section (pp. 20 – 21):

One question that remains following our study concerns the important transition between anticipatory states and the earliest moments of the sensory response that are essential to rapid and accurate target detection. We supposed that the strength of the population response during the initial onset transient was an important component of this process, and found that anticipatory states were related to the strength of the population onset transient response to non-target stimuli (Figure 8). However, to directly determine which aspects of the population sensory response link anticipation to target detection, it would be necessary to analyze the sensory responses to target stimuli. Because the total number of stimuli that can be presented in an experimental session is limited, there is an inherent trade off in the design of attention experiments between the number of non-target and target stimuli shown. In this study, we favored a large proportion of non-target stimuli, at the expense of fewer targets. The advantage of this design is that we had many repeated presentations of physically identical stimuli uncontaminated by motor responses, but the drawback of this design is that we had too few presentations of any given target stimulus to directly examine which features of population responses were essential for target detection. A future study utilizing a different proportion of target to non-target stimuli would help to answer this important question.

To address the question posed about population activity, we examined the subpopulation responses to non-target stimuli that the reviewer suggested (Supplemental Figure 1). To our eyes, the striking feature is that there was very little attention effect during the initial transient response across all of the subpopulations. Investigations targeting how attention modulates the transient response are indeed a critical area of future inquiry, and we hope our findings here provide some direction for those studies, particularly regarding the use of population codes during anticipatory periods.

Supplemental Figure 1: Peristimulus spike time histograms (PSTHs) for subpopulations of neurons selected for the pattern of their attention effects. Left: neurons that were significantly suppressed with attention pre-stimulus and significantly enhanced with attention post-stimulus. Middle: neurons that were significantly enhanced pre-stimulus and were not significantly suppressed post-stimulus. Right: neurons that did not show an attention effect pre-stimulus and did show an attention effect (in either direction) post-stimulus. Notably, for all subgroups of neurons the time-course of attention effects (bottom row) follows a gradual ramp after stimulus onset, without large attention effects during the onset transient response.

Moreover, modulations may be different in specific ways for the target stimulus, especially in populations best suited to detect subtle orientation changes, ie. those where the steepest part of the tuning curve falls at the sample-target orientation difference. A way to address this might be analysis of responses to the target across all neurons and across those to which the previous applies. While this can be affected by the saccade, it should be possible to analyse at least the first 150 ms.

This is a very interesting point (raised also by Reviewer 3) and one that we tried to examine. In designing our study, we opted to have a greater proportion of non-target stimuli, since these were by definition physically identical and would provide many repeated presentations for signal averaging. This preponderance of non-target stimuli necessarily came at the cost of few target presentations, as we mentioned above. For reference, Monkey P was presented with 11.49 (3.14) [mean (SD), across

sessions] valid targets of a given orientation (of 8 possible target orientations) at each location, and Monkey W was presented with 14.56 (4.75) valid targets of a given orientation at each location (now mentioned in the revision on page 24). Combined with the contaminating effect of eye movements that the reviewer mentions, we found we were simply unable to derive reliable estimates of population responses to target stimuli. We agree with the reviewer that our results suggest interesting hypotheses about population responses to targets that could potentially be addressed in a future study designed with these issues in mind. We now discuss this possibility in the Discussion section of the revision (pp. 20 – 21, as also quoted in response to the previous point).

The question of how the magnitude of attention effects might be related to the orientation preferences of neurons is an interesting and important question (raised also by Reviewer 3), and one that we tried to examine in a preliminary analysis. At the beginning of each session, we ran a quick experiment in which the animals passively fixated a dot while we showed briefly flashed gratings in the RF area to estimate orientation tuning curves (20 repeats at 18 orientations). During our preliminary analysis we looked to see if the attention effect of a given neuron depended on the sample stimulus for the trial (i.e., 45 vs. 135 degree orientation), relative to the neuron's orientation preference. We did not see evidence for such a relationship from that preliminary analysis (see figure below). One potential issue is that the spatial and temporal frequencies and size of our stimuli were not optimized to drive any particular neuron, making it difficult to derive reliable estimates of tuning curves based on this quick experiment at the beginning of each day. A future study that spends more time mapping out the response properties of individual neurons could better address this interesting question, but such detailed mapping would also necessarily reduce the time that could be spent on the attention experiment each day. We have revised the report to discuss this important issue of how attention effects might relate to feature tuning of neurons (p. 19):

Another important source of diversity across neurons is tuning to stimulus properties. Because our task involved discriminating a change in orientation, feature-based attentional strategies could have impacted how the responses of neurons were modulated. For example, neurons preferring orientations at, or near, the orientation of the stimulus at the attended location might exhibit greater enhancement with attention than neurons preferring orientations quite different from that of the attended stimulus⁴³. Such interactions between spatial and feature-based attention during stimulus responses have been the subject of substantial previous research⁴⁴⁻⁴⁹. Our results indicate that sensory context (i.e., anticipatory vs. stimulus-processing states) is another important contributor to diversity in population effects of attention. One drawback of our study design, which used sample stimuli of only two orientations, is that we did not have a compendium of stimuli sufficiently varied to assay how anticipatory attention states interact with stimulus features. Moreover, because the direction and magnitude of the orientation change was unpredictable for our task, a feature-based attentional strategy may have been less fruitful than if a predictable orientation change was used. A future study utilizing a richer stimulus set could help to address this important question regarding anticipatory attention states in neural populations.

For each session, we estimated each neuron's preferred orientation and attention modulation index (pre-stimulus). We measured attention modulation when the sample stimulus in the RF was at 45 degrees (top) or at 135 degrees (bottom). For each orientation condition we fit the following equation to the data:

$$A = \alpha \cos(2\pi(x - \theta)) + c$$

Here, 'A' is the attention modulation index, 'x' is the preferred phase of a neuron, 'θ' is the preferred orientation with the peak attention effect, and 'c' is a constant. The plot above shows the distribution of fitted 'θ' values across all sessions. One prediction is that attention effects might be greater for neurons that prefer orientations around the orientation of the stimulus in their RF. If that hypothesis were true, then we might have observed some consistency across sessions for the preferred orientation with the peak attention effects. Instead, we see a distribution that is fairly uniform, which is consistent with the null hypothesis that there is no relationship between preferred orientation and the magnitude of attentional modulation. This null result could be related to imperfect estimation of orientation tuning curves for individual neurons. It may also be that our task, in which the potential target stimuli on each trial could have a wide range of orientations, did not lend itself to a strategy in which attention is directed to neurons with particular orientation preferences.

2. The authors argue that this mixed gain change, can disentangle the effects of attention and the effect of a weak stimulus being presented, by ensuring that some cells take on negative weights during anticipatory attention in downstream areas. I think this needs at least clarification in terms of how the authors think that is implemented, and how a negative weight could be switched to a positive weight after stimulus presentation?

In our discussion we entertained a few possible mechanisms for this pattern of results (p. 18), including a simple computational model that reproduced our findings qualitatively (Supplemental Figure 4). This model relied on differential population coupling across the neurons in a brain area, a connectivity motif that has been reported on previously (Okun et al., 2015, *Nature*). The model is a system of two differential equations, each representing the firing rate of a model neuron. The firing rate of one model neuron depends on the amount of stimulus drive, multiplied by an attention gain factor. The firing rate of the second neuron is determined by comparison of the firing rate of the first neuron to some

background activity level, and does not depend directly on the stimulus or the attention factor. The first model neuron is driven strongly by the stimulus input, while the second model neuron has strong coupling to the local population (this is comparable to the “soloist” and “chorister” framework of Okun et al.). This minimal model of two differential equations is sufficient to replicate the qualitative pattern of our results, and shows how diverse connectivity in a neural population could serve as a simple mechanism. Since our model is speculative in nature, we believe that the right place for it is in the supplemental materials. However, we have revised our discussion to draw increased attention to our thinking about potential implementations for the results that we observed (pp. 18 – 19):

It is well-appreciated that the neurons in a cortical area have diverse functional roles. Research classifying neurons based on their primary neurotransmitter and morphological properties has revealed that modulatory influences can depend on the state of ongoing activity^{40, 41}. Thus, cell class diversity in the population could provide one potential explanation for the diversity of attention effects between the stimulated and unstimulated state. Even within a single class of cells, distinct functional roles for neurons can be conferred based on network topology. For example, neurons within V4 vary in the degree of their coupling to the local population, and neurons with weak population coupling are modulated more by sensory factors than neurons with strong population coupling⁴². We used a simple computational model with two neurons to test whether differential population coupling could provide another potential explanation for the results we observed. One model neuron’s firing rate depended only on stimulus drive and an attention gain factor, reflecting weak population coupling; the other model neuron’s firing rate depended only on the rate of the first neuron, reflecting strong population coupling (Supplemental Figure 4). This minimal model was sufficient to replicate the qualitative pattern of results between the stimulated and unstimulated states, and illustrates that differential influence by local or distant inputs could provide a mechanism for diverse manifestations of attention modulation in a population.

Supplemental Figure 4: Simple dynamical system model consistent with the pattern of results that we observed in the neural population data. A) system of equations governing the dynamical system. The firing rate of one neuron (n_1) depends on stimulus drive (s), which is multiplicatively scaled by an attention factor (a). The firing rate of the other neuron (n_2) depends only on the firing rate of n_1 , relative to a constant value ($b = 5$). This latter non-linear “comparison” operation could be achieved through recruitment of local inhibitory interneurons. B) Conceptual schematic of the model. Top-down endogenous attention signals (red) interact with bottom-up stimulus drive (brown) to excite n_1 (blue), which in turn modulates the firing rate of n_2 (green), with additional involvement of local inhibitory interneurons (orange). C) Time series of firing rates for model neuron n_1 with high attention ($a = 2$, red) and with low attention ($a = 1$, blue). The neuron is facilitated by attention in both the absence ($s = 0$) and the presence ($s = 10$) of a stimulus. D) Time series of firing rates for model neuron n_2 . This neuron is also facilitated by attention in the presence of a stimulus, but is suppressed by attention in the absence of a stimulus. E) Time series of the population average firing rate for the two model neurons. The population shows a typical gain-like effect in response to a stimulus, but no net effect of attention in the absence of a stimulus because of the offsetting effects of attention on the two neurons. Thus, this minimal model demonstrates that heterogeneous population coupling could provide one potential mechanistic explanation for the pattern of results that we observed for the neural populations, although it does not rule out additional or alternative mechanisms.

3. I think a real statistical problem is to treat units recorded across session as independent observations. Given the Utah array recordings, this is extremely unlikely to be the case, and thus re-sampling is a real issue, and the implications on statistics. This is most prevalent when the effects are calculated for a

session (e.g. attention axis), and then the p-value is based on the number of session. Assume (worst case scenario), the same neurons were recorded every session. Then the same effect would have to occur from session to session, and the p-values would be meaningless. I think it is important to establish, to what extent neurons differ between sessions, or think about different ways to address this problem.

The reviewer raises an important issue, and one that we've thought about deeply. We endeavored for the revision to quantify the amount of turnover on our arrays as the reviewer suggested (Supplemental Figure 6). We found evidence that a substantial proportion (typically around 50%) of our sample changed between consecutive sessions, based on a waveform shape similarity metric. Considering the consistency of our results across sessions (e.g., Figure 6d) despite this high turnover rate on our arrays (and also replicated in two animals), we feel it is unlikely that this consistency is due only to repeatedly observing a static sample. This reasoning is now mentioned on page 23 of the revision.

Supplemental Figure 6: the recorded sample of neurons changed substantially between sessions. We considered each session in sequence and determined the edit distance for the sample from the previous session (i.e., the number of neurons that were added or lost from the previous session). We determined that a neuron was plausibly retained across sessions if there was a neuron on the same channel with a waveform shape that was 95% correlated with a neuron on the same channel the session before. We normalized the edit distance as a proportion of the sample size of the earlier session. That is, a value of 1 means that if the sample on one day was 100 neurons, then a total of 100 neurons were either lost or gained on the next day. The value could be greater or less than one, which would simply mean the amount of neurons lost and gained was greater or less than the sample size on the earlier day in the pair, respectively. Typically, about half the sample turned over on consecutive days. This turn-over ratio was greater if there was more time between consecutive recording sessions. Given this substantial variability across recording sessions, it is likely the consistency of results that we observed over time (e.g., main text Figure 6) reflects a general property of V4 neurons, and not any one particular sample of neurons.

The reviewer is correct in that if we had, in fact, repeatedly sampled the same pool of neurons across sessions, then the p-values would have a different meaning than if we had sampled a completely new set each time. In the former case, the interpretation would be that a particular sample of neurons in V4 shows the effects we found (and that that effect was not due to chance, e.g. on the first session), whereas in the latter case the interpretation would be that V4 neurons in general show the effect we found. Both are valid inferences, and both are interesting findings. In fact, our view is that attention mechanisms can be conceived as latent factors (represented by our attention axes) that affect V4 populations, and that we are observing the same latent factors each session with our population-level analysis whether or not the neurons are the same or completely different between sessions. This view is not only our own, but is shared by other research groups (e.g., Cohen and Maunsell, 2010, *J Neurosci*; Mayo et al., 2015, *PLoS ONE*). Therefore, our main conclusion holds (i.e., that a different mixture of neurons in V4 characterizes attention effects during anticipatory periods than the mixture of neurons that characterizes attention effects during stimulus processing) regardless of which interpretation of the significant statistical result is embraced.

Minor:

4. The authors argue that the unattended hemisphere has knowledge about the attentional state of the attended hemisphere. This is maybe too strong an argument. If attend away leads to a different state than no attention, and yet then attend to, the finding that the neurons can decode between hits and misses in the unattended location, may simply be a result of attentional lapse (no attention). And this would possibly be expected to be less pronounced than differences between attend RF and misses, if attentional 'down modulation' during attend away was less pronounced than up modulation in attend RF.

We agree with the reviewer that our finding does not rule out the possibility of a non-spatially-selective shift in attentional state that includes the ipsilateral hemifield (which would be, after all, information about the attentional state everywhere, including ipsilaterally). We now explicitly address this potential interpretation in the manuscript (p. 11). However, we also reason that the result cannot be understood completely in terms of attentional lapses, since we found that when attention axes projections were strongly indicative of attention to the RF location (i.e., not lapsed), performance was relatively impaired for targets in the ipsilateral hemifield, suggesting a withdrawal of attentional resources from that location. We also now discuss this interpretation of the results (p. 11):

The corresponding pattern was seen for targets out of the RF with the signs reversed, albeit less consistently (Figure 6e). This latter result is surprising, because it indicates that the V4 neurons accessed by our array in only one hemisphere carried information about the attention state of the subject with respect to locations in the ipsilateral hemifield. This could be consistent with a competition for processing resources between the two hemispheres (i.e., strong attention to the RF implies a withdrawal of processing resources for ipsilateral space³⁵). Alternatively, our procedure to find the attention axis for targets in ipsilateral space may tap into non-spatially specific attentional processes, such as arousal.

5. Typo in methods, line 605: If the both the....

Fixed. Thank you.

Reviewer #2 (Remarks to the Author):

This paper examines the neural correlates of “anticipatory attention” in the dorsolateral prefrontal cortex (dPFC), during an epoch in a change detection task in which the attentional locus is cued but before a stimulus appears. The results show that the locus of attention can be read out from the population response in dPFC using linear decoding, and that this readout is independent of the gain changes that are observed after stimulus presentation.

Our recordings were actually in V4, not dPFC. We have now added a reference to this in the abstract to ensure it is as prominently available as possible in the paper, in addition to the other locations in which it is stated.

The paper is potentially interesting, but there are several major deficiencies in the analysis that make the results questionable.

Some of the issues raised by this and other reviewers were likely due to suboptimal communication of our analyses and results. We have taken advantage of these opportunities to improve the communication of our study, and we believe the revised manuscript has benefited as a result of this feedback.

1. The main argument of the paper is that there are population effects in the pre-stimulus response that are orthogonal to those in the post-stimulus response and capture some of the behavioral variance. However, throughout the paper we are never told *how much* variance – neural and behavioral – is captured by these components. Without this information it is very difficult to estimate the significance of the results.

By construction, the attention axes are those directions in the population space that maximize the between-attention-condition variance in the population responses. Although maximal, the effects of attention on stimulus responses are not always large in terms of percentages. This is particularly true when stimuli are not optimized to best drive neurons being tested, and there is only one stimulus presented in the neurons’ receptive fields. The neural effect sizes of attention on stimulus response that we observed are typical for studies similar to ours. For example, when using sequentially presented stimuli such as ours, Luck et al. (1997, *J Neurophysiol*) found an average attention gain factor on stimulus responses in V4 of 6%. Similarly, when using a task nearly identical to ours, Cohen and Maunsell (2009, *Nat Neurosci*) reported an average attention gain factor of 8-10%. Our reported average for the gain in stimulus responses (9.2%, Figure 4), falls squarely in line with these highly-cited prior reports, and we now explicitly mention this correspondence in the revised manuscript (p. 8):

Specifically, the mean increase in sustained, stimulus-evoked firing rate was 9.2% (Figure 4c), which is comparable to the findings of Cohen and Maunsell (2009), who found gains of 8-10% using a very similar task.

Although attention effects on firing rates may not be large in terms of percentages, it is an open question the degree to which a few percentage points of change in firing rates modulate perceptual processes. It is important to consider how the activity is read out by downstream populations guiding behavior, which is precisely what motivated our study. With respect to effect sizes for behavior, Figure 7

of our report illustrates that the attention axes account for a relative swing in hit rate of approximately 10% (pre-stimulus attention axis) to 20% (post-stimulus attention axis).

2. This concern is highlighted by the only mention of percent variance explained (PEV) that comes in the last figure (no figure numbers are provided). That figure is inscrutable (very difficult to follow, along the lines described below) and the axes only go up to 3% of total variance explained (up to 0.006 of the residual variance) which seems like a ridiculously small number.

The effect here, although admittedly small in terms of percentages, is statistically reliable. Since we are recording only a miniscule fraction of all the neurons in V4, and there are many brain regions outside of V4 involved in attention, it is difficult to know how to judge whether the absolute size of a neural effect is big or small. The main point of Figure 8 is to illustrate the relative *timing* at which each attention axis provides reliable information about the vigor of response to a subsequent stimulus, rather than the magnitude of that information. We have revised the manuscript to highlight this particular emphasis (pp. 14 – 15; figure legend on p. 45; additional annotations to Figure 8):

We therefore assessed how different attention signals measured from the population affected the V4 population activity at different times during the response to a stimulus. The key prediction is that if anticipatory attention is particularly important for target detection then anticipatory attention states should be particularly informative about the earliest part of the sensory response, rather than the later part of the sensory response after a decision has already been made. To test this, we used the attention axes projections described above to estimate the attention state on each trial. Then we asked how much trial-to-trial variance in population stimulus responses we could explain based on those attention estimates. In particular, we were interested in identifying the time during the response at which any such relationship would be strongest. Because we predicted the population activity pattern and not the overall spike rate averaged across the population, and because we used a continuously variable estimate of attention state rather than a binary division between the two cue conditions, this analysis did not have to show the same time-course as the attention modulation of the PSTH (Figure 2).

We began this analysis by using the post-stimulus attention state to predict the population response to the subsequent stimulus, and we found that we were best able to predict spiking activity around 300 ms after the stimulus onset (Figure 8, brown). This time course is consistent with the PSTH results (Figure 2), although we noticed that prediction of the early-latency V4 responses (~50 ms after stimulus onset; Figure 8, brown) was improved by these trial-by-trial attention estimates compared to simply averaging trials over an entire block for each attention condition. Critically, we found that we could account for yet more variability in stimulus responses if we included the pre-stimulus attention state as a predictor variable (Figure 8, orange). Our aim with this analysis was to compare the timing of when each estimate of attention state provided the most reliable information about subsequent stimulus responses, rather than the absolute strength of that information. In line with our guiding thesis, the additional predictive power of the pre-stimulus attention state peaked early in the visual response, around 71 ms (Figure 8, orange). This suggests the unique features of population activity patterns during anticipatory attention reflect mechanisms enabling a robust sensory response on the rapid time scales that support brisk and accurate discrimination.

3. The analysis apparently uses multi unit and single unit data, making the conclusions highly questionable, especially given the very small effect sizes.

To address this, we re-performed our analyses while restricting to neurons with SNR > 3 (29.0 ± 7.5 units per session for Monkey P; 11.2 ± 5.6 units per session for Monkey W), and found that the overall pattern of results was similar to what we observed for the full populations. This now appears as a new Supplemental Figure 5 (p. 22).

Supplemental Figure 5: Main analyses performed including only well-isolated units (putative single neurons); related to article Figures 6 and 7. We quantified recording isolation by computing signal-to-noise ratios (SNRs) for each unit action potential waveform (defined as the average amplitude of the waveform divided by the standard deviation of the waveform noise), and included only units with $SNR > 3$. The pattern of results for this restricted analysis is similar to what we found when we did not exclude any units on the basis of SNR, as reported in the main article.

The attention axis represents a weighted mixture of neural responses across the population. Therefore, there is some question as to the benefit that would be conferred by separating neural responses attributable to single units simply to mix them together again for the population-level analysis. We thought deeply about the concern raised by the reviewer and were unable to reason an explanation for how including multi-unit activity in our analysis could have *created* the result that we found. Indeed, we reasoned that, if anything, including multi-unit activity could have only reduced our effect sizes; therefore including them is the conservative approach (it is also what was done for the most relevant, previous studies: Cohen and Maunsell, 2010, *J Neurosci*; Mayo et al., 2015, *PLoS ONE*; which we now mention in the revised methods section on page 22). We do show individual neuron responses in Figure 3. We've now added the signal-to-noise ratios (SNR; defined as the ratio of the average action potential waveform amplitude to the standard deviation of the waveform noise [described on page 22 of the revision]) for the action potential waveforms of the illustrated units to the figure so that readers will better be able interpret the isolation of these units (minimum SNR in the figure is 2.3). We highlighted these examples because the change in direction of the attention effect suggests a strong and intuitive refutation of a stimulation-invariant gain mechanism at the scale of individual neurons. However, our broader conclusion concerns a substantial difference in the pattern of attention modulations at the population level, and this is best reflected in our main, population-level analysis.

4. The penultimate figure shows correlation between neural projections and hit rates, but this is confounded with response bias. The appropriate analysis should use d' .

The reviewer raises an important point. In most studies of attention, improvements in hit rate can be attributed to both shifts of criterion and sensitivity (d'), and both such effects are typically observed. It can be important to consider how each of these components of attention might be reflected in neural activity. A recent excellent study by Luo and Maunsell (2015, *Neuron*) examined this question in detail. They designed their experiment to cause animals to exclusively shift only criterion or sensitivity with the focus of their attention and found that neural correlates of attention in V4 were only related to sensitivity shifts. To confirm that the neural effects that we observed were also related to sensitivity shifts, we performed the analysis on d' values that the reviewer suggested and included the results as a new supplemental figure (Supplemental Figure 2; p. 12). We found that d' varied systematically with shifts along either attention axis similarly to the hit rate effects that we reported. This confirms that the neural effects we observed are related to changes in sensitivity.

Nevertheless, we believe it is important to report hit rate effects in the main article because this facilitates comparison with the two most-related previous studies in the literature, which both reported hit rate effects (Cohen and Maunsell, 2010, *J Neurosci*; Mayo et al., 2015, *PLoS ONE*).

5. The authors use t-tests to look at firing rate differences without even a mention of whether they did normality tests. This use must be justified or replaced with non-parametric tests. Compounding these concerns, the writing suffers from a pervasive lack of clarity that makes the paper very frustrating to read and understand.

We tested for violations of normality assumptions using Lilliefors' test where t-tests were used. We have added mentions of this step in the appropriate sections of the methods (pp. 26, 28, 29, 32). The only analysis where violations of the normality assumption were found was the test for significant attention effects on the responses of individual neurons in Figure 4. We therefore re-performed this analysis using

non-parametric Mann-Whitney U tests instead of t-tests. This change did not meaningfully alter the pattern of results or our conclusions. Some individual neurons that were not significant by the t-test were found to be significant by the Mann-Whitney U test, and vice-versa. We updated the reporting of the number of neurons with significant effects in the results section accordingly (p. 8; Figure 4). The example neurons selected for illustration in Figure 3 were all significant by either test.

1. The task and analyses are carelessly described. Monkeys are trained on a task in which, in alternating trial blocks, they must attend to a Gabor stimulus in the right or left hemifield and make a saccade when they detect a change in this stimulus. Stimuli flash on and off (400 ms on, 300-400 ms off) and the change has a 40% probability of happening on each cycle after the initial “sample”. The focus of the paper is on activity before stimulus onset – during the “off” period – but throughout the paper there is a lot of confusion about which one of the several off periods, this analysis is applied to. First, it would help to have a histogram of the number of cycles before a response across all the trials in their data set. Second, the authors should be explicit about which one(s) of the off epochs they include. Third, they should discuss – and rule out – any effects of hazard rate – the monkeys’ increasing anticipation of a change as time progresses. The latter point is very important and may change both behavior and neural responses. Fourth, they should discuss whether they include valid and invalid trials since the saccade direction differs in these cases (if both trials are correct).

Targets were presented at a uniform hazard rate (p. 25), which corresponds to an exponential distribution of sequence length. We have added a supplemental figure showing distributions of sequence length, and the subjects’ performance for both valid and invalid targets at near-threshold difficulty as a function of sequence position (Supplemental Figure 7). Performance was consistent for sequences up to ~6 stimuli; performance at sequences longer than 6 stimuli was difficult to estimate reliably because of their infrequent occurrence.

Supplemental Figure 7: Behavioral performance was consistent as a function of sequence length. Target

probability followed a uniform hazard function, resulting in an exponential distribution of sequence lengths. For sequence lengths that occurred frequently enough to reliably estimate discriminability (i.e., sequences of up to 6 stimuli), behavioral performance did not vary as a function of sequence length.

We have made changes to the methods to enhance the emphasis as to which inter-stimulus intervals were used for analysis (pp. 27 – 29). In particular, we used the inter-stimulus interval immediately preceding target onset for both defining the pre-stimulus attention axis and for analyzing the relationship between neural activity and behavior. This latter point is also stated in the legend for Figure 6.

We defined attention axes using population activity during two epochs: 1) a pre-stimulus attention axis defined using the responses during the 200 ms immediately preceding target onset, and 2) a post-stimulus attention axis defined using the responses from 200 to 400 ms following the onset of the stimulus that preceded the target. We always projected data onto the axis defined during the corresponding time period of interest (i.e., pre-stimulus data onto the pre-stimulus attention axis and post-stimulus data onto the post-stimulus attention axis). (p. 28)

Valid and invalid trials were both used, but no data from the time of target onset through the end of the trial (including saccades) were used at any point. We have clarified this on p. 25.

2. Confusion is particularly high in the figure comparing hit and miss trials (there are no figure numbers). That analysis suggests that the monkey makes similar number of hits and miss trials in a session. How does this reconcile with the high d' values in Fig. 1? Are these trials selected in some way? And what is the “pre-stimulus” epoch and the “pre-target epoch”? How do these relate to the epochs in the previous figures? And which attention axes are used to calculate the projections? Is pre-stimulus activity (or pre-target activity) projected on the post-target axes? (and if so, what is the justification, given that we are interested in the pre- and post-target components that are orthogonal to each other?)

Our behavioral results show that d' is only high for three of the five target conditions (Figure 1). Performance was poor for orientation changes of at most 3 degrees, and even worse for invalid targets. Therefore it is not surprising that subjects missed a substantial number of targets as is illustrated in Figure 6. All of the available trials are included in the analysis.

With respect to which data are projected onto which attention axes, we have added emphasis on page 28 of the revision that pre-stimulus activity was always projected on the pre-stimulus attention axis and post-stimulus activity on the post-stimulus axis. By “pre-target” epoch, we meant the epoch of spontaneous activity preceding a target onset (i.e., a stimulus with an orientation change). However, to avoid confusion we have changed this terminology in the revision to “interval preceding a target”. It is also specified in the legend to Figure 6 that we “...projected the high-dimensional population activity from 200 to 400 ms following the onset of the sample stimulus that preceded each target onto the post-stimulus attention axis...” and then we “...[projected] the spontaneous population activity during the 200 ms immediately preceding the onset of each target onto the pre-stimulus attention axis...”. This is illustrated graphically at the top of Figure 6.

3. Similar concerns – too many to list – come up for the remaining figures.

We have made clarifying changes throughout the manuscript, including to figures and figure legends.

Reviewer #3 (Remarks to the Author):

Substantial work has demonstrated that selectively attending to one of several objects will increase the firing rate of neurons tuned to the location of the objects within the visual field. In some studies (almost always in fMRI, sometimes in EEG/MEG, rarely in electrophysiology), changes in pre-stimulus baseline activity levels is also observed – typically such that greater activity precedes the presentation of a stimulus in an attended part of space. But, given that these effects are challenging to see in electrophysiological recordings, and are often obscured when averaging across all recorded neurons, perhaps the typical story that attention induces a gain in responses, regardless of stimulation, is incomplete.

The authors found striking evidence for an alternative model that allows for different types of modulation (i.e., suppression vs enhancement) at different phases of the trial, within the same individual neuron. Such results are demonstrated to be unlikely to occur due to a simple stimulation-independent gain model, are related to behavioral performance, and allow for independent readout of pre-stimulus attentional state and post-stimulus attentional enhancement of stimulus representations.

I think this is a really excellent report which the readership of Nature Communications will find fascinating. The question is novel and important, the analyses employed are elegant, the data is presented thoroughly, and the writing is exceptionally clear. In particular I very much appreciate Figure 4, which shows the full set of modulations across the dataset. These figures are somewhat rare in the literature, but are also so important for getting a handle on the true expected distribution of these effects.

It's always a pleasure to review a report that is so carefully and thoughtfully prepared. I have a few conceptual questions, mostly out of my own curiosity, and a couple of very minor suggestions that could improve the presentation of the results. Perhaps the authors may be interested in adjusting the manuscript to include some of the conceptual issues I raise, but I emphasize that this would be purely optional and the choice not to include such text would not in any way impact my judgment of a revision, should that be necessary.

We are grateful for the reviewer's kind words. The questions below are important ones that we've thought about as well.

1. The authors estimated RFs/tuning functions for each neuron. The differences in orientation the monkeys were required to detect were quite small – this suggests that the animals could perform better if they preferentially allocated gain to neurons tuned slightly away from the non-target orientation values (see, for one of many examples in the human literature, Scolari et al, 2012, J Neuro). Without emphasizing this particular possibility too much, I'm curious if the authors have evaluated whether orientation preferences modulate the attention gain effects they see (either pre or post stimulus)? This is already quite a comprehensive manuscript which sufficiently makes its key points, so I leave it to the authors' discretion whether to explore and/or discuss such a possibility.

How the magnitude of attention effects on firing rates might be related to the feature tuning preferences of individual neurons is an interesting and important question, and one that was raised by Reviewer 1 as well (point 1). At a preliminary stage of analysis, we did look into this question. We did not

find evidence for a relationship between attention effects and orientation tuning preferences during that analysis (see figure below).

For each session, we estimated each neuron's preferred orientation and attention modulation index (pre-stimulus). We measured attention modulation when the sample stimulus in the RF was at 45 degrees (top) or at 135 degrees (bottom). For each orientation condition we fit the following equation to the data:

$$A = \alpha \cos(2\pi(x - \theta)) + c$$

Here, 'A' is the attention modulation index, 'x' is the preferred phase of a neuron, 'θ' is the preferred orientation with the peak attention effect, and 'c' is a constant. The plot above shows the distribution of fitted 'θ' values across all sessions. One prediction is that attention effects might be greater for neurons that prefer orientations around the orientation of the stimulus in their RF. If that hypothesis were true, then we might have observed some consistency across sessions for the preferred orientation with the peak attention effects. Instead, we see a distribution that is fairly uniform, which is consistent with the null hypothesis that there is no relationship between preferred orientation and the magnitude of attentional modulation. This null result could be related to imperfect estimation of orientation tuning curves for individual neurons. It may also be that our task, in which the potential target stimuli on each trial could have a wide range of orientations, did not lend itself to a strategy in which attention is directed to neurons with particular orientation preferences.

However, in a variety of ways, our study was not optimized to tackle that question head on, and it would be better tackled in future work. For example, because our stimuli were not optimized to drive individual neurons, estimating orientation tuning curves was difficult. Also, the direction and magnitude of orientation changes were unpredictable, which may have undercut potential feature-based selection strategies (an effect that was also reported by Scolari et al. when target features were unpredictable vs. predictable). We have added discussion of these issues to the revised manuscript, including a reference to the suggested prior report (p. 19):

Another important source of diversity across neurons is tuning to stimulus properties. Because our task involved discriminating a change in orientation, feature-based

attentional strategies could have impacted how the responses of neurons were modulated. For example, neurons preferring orientations at, or near, the orientation of the stimulus at the attended location might exhibit greater enhancement with attention than neurons preferring orientations quite different from that of the attended stimulus⁴³. Such interactions between spatial and feature-based attention during stimulus responses have been the subject of substantial previous research⁴⁴⁻⁴⁹. Our results indicate that sensory context (i.e., anticipatory vs. stimulus-processing states) is another important contributor to diversity in population effects of attention. One drawback of our study design, which used sample stimuli of only two orientations, is that we did not have a compendium of stimuli sufficiently varied to assay how anticipatory attention states interact with stimulus features. Moreover, because the direction and magnitude of the orientation change was unpredictable for our task, a feature-based attentional strategy may have been less fruitful than if a predictable orientation change was used. A future study utilizing a richer stimulus set could help to address this important question regarding anticipatory attention states in neural populations.

2. Related – a pure gain increase may not optimally enhance the amount of information about the relevant sensory variable (orientation). Perhaps the authors might care to discuss the relationship between response intensity/spike rate and information content of the population code, and how attention can impact one, the other, or both? Was this issue examined in the present dataset? I imagine the task design makes this somewhat challenging – since the target stimuli necessarily result in a very fast oculomotor response, all analyses must be focused on the samples. In any case, I'm curious what the authors think about this. A potentially relevant report in humans is Myers et al, 2015, eLife.

This is another critical question, and one that was alluded to also by the first reviewer. Unfortunately, we did not have enough repeated presentations of each target stimulus to properly address this issue (~10-15 per condition each session; p. 25). At a preliminary stage we looked at how well a decoder could disambiguate the two non-target orientations, but ran into issues of ceiling decoding performance (they were separated by 90 degrees), so attention had little room to improve decoding of non-targets. A future study that had a higher proportion of target stimuli (coming with the necessary expense of fewer repeated presentations within each condition) would be needed to tackle this important question. In that case a decoder could be trained directly on the more challenging target orientation changes. Although we were unable to address this issue fully in our study, we agree with the reviewer that it merits further discussion in our manuscript. We now discuss the issue of target processing on pages 20 – 21 of the revised report:

One question that remains following our study concerns the important transition between anticipatory states and the earliest moments of the sensory response that are essential to rapid and accurate target detection. We supposed that the vigor of the population response during the initial onset transient was an important component of this process, and found that anticipatory states were related to the strength of the population onset transient response to non-target stimuli. However, to directly determine which aspects of the population sensory response link anticipation to target detection, it would be necessary to analyze the sensory responses to target stimuli. Because the total number of stimuli that can be presented in an experimental session is limited, there is an inherent trade off in the design of attention experiments between the number of non-target and target stimuli shown. In this study, we favored a large

proportion of non-target stimuli, at the expense of fewer targets. The advantage of this design is that we had many repeated presentations of physically identical stimuli uncontaminated by motor responses, but the drawback of this design is that we had too few presentations of any given target stimulus to directly examine which features of population responses were essential for target detection. A future study utilizing a different proportion of target to non-target stimuli would help to answer this important question.

3. Finally, one last conceptual issue related to orientation preference: does the pre-stimulus change in firing rate depend on orientation tuning and the 'sample' orientation within the RF on that trial? If so, this could speak to a particularly strategic re-apportioning of metabolic resources by attention on a trial-by-trial basis.

This is related to point 1 above. Again, we examined this during a preliminary stage, but don't have enough confidence in our estimates of preferred orientation to address this question with the current dataset. For the figure above in response to point 1, we separated trials based on whether the stimulus in the RF was 45 degrees or 135 degrees (spontaneous data are shown, but after the subject knows what the 'sample' orientation is for the trial). Then, for each neuron, we measured the attention modulation when the RF stimulus was attended vs. unattended. We also tried to estimate the orientation preference of each neuron (as best we could). We asked if the strength of modulation varied as a function of orientation preference, depending on the orientation of the stimulus shown in the RF (tested by fitting a cosine function to the data and taking the position of the peak of that function). We did not see evidence that the strength of attention modulation was related to neurons' orientation preferences, regardless of which orientation was the 'sample'. There are many possible explanations for this null result. One is that the unpredictable nature of the direction and magnitude of the orientation change that signaled a target might have undercut the utility of a feature-based strategy. Another highly likely explanation is that we were unable to accurately estimate neurons' orientation preferences because we optimized our study for different scientific goals.

For the main study, we separately performed our analysis for each sample orientation. Attention axes were different for the two sample orientations, but not in any obviously systematic way across sessions, as suggested by the uniform distributions shown in the above figure.

4. In Figure 4 – do the brown/orange arrowheads indicate the mean of the full distribution, or the mean of the significant neurons within each marginal distribution?

Mean of full distribution. We now specify this in the legend for Figure 4.

5. Figure 6 – the small insets in panels b & c – what is the x range on each of those? -1 to +1?

To provide a sense of scale to the inset plots (Fig 6b,c) without unduly cluttering the figure, we have added labels for the means of each distribution shown, which are the most relevant quantities. The attention axis projection values are normalized with the aim of setting the mean of each "hit trial" distribution at +1 (for the RF targets as shown), but that goal isn't exactly achieved because cross-validation is used.

6. I'm curious how orthogonal the pre- and post-stimulus attention axes are pre-orthogonalization? (what is their vector angle?).

About 75 (Monkey P) to 80 (Monkey W) degrees. We have added this detail to the manuscript (p. 29).

7. Figure 5a – I'm not sure the 2-d case is the most optimal to show since there is only a single possible orthogonalized pre-stimulus attention axis for any 2d post-stimulus attention axis. Would a 3d example be possible? In this case (if I'm understanding correctly), the pre-stimulus attention axis would still lie along the plane defined by the post-stimulus attention axis and the 'raw' pre-stimulus attention axis (gray arrow), but will be oriented orthogonal within that plane. That is – the gray arrow restricts which of the infinite possible orthogonal vectors is called the pre-stimulus attention axis. (and I do apologize if I'm a bit lost confused here – I'm happy to be corrected by the authors)

The reviewer is describing this correctly. As the reviewer states, there is only one axis in two-dimensional space that is orthogonal to the post-stimulus attention axis, but in higher dimensions there are infinitely many, so finding the best orthogonal axis is not trivial. We tested a 3D version of this figure, but found the improvement in technical accuracy was offset by a detriment of intuitive clarity, particularly for readers not already familiar with high-dimensional population analyses. To address this concern while keeping the figure as clear as possible, we've added a remark to the figure legend (pp. 28 – 29) to the reviewer's point that in >3 dimensions there are infinitely many orthogonal axes, and finding the best of these is not trivial:

...although there is trivially only one axis orthogonal to the post-stimulus attention axis when $N = 2$, as in Figure 5, in general there are infinitely many orthogonal axes when $N > 2$, so finding the best orthogonal pre-stimulus attention axis is not trivial.

8. The authors may wish to add a code/data availability statement. I would encourage they publicly deposit data/code in an easily-accessible permanent repository like Open Science Framework (osf.io)

We have added a code/data availability statement on p. 33:

Analysis computer code and data for this project are available from the authors upon request.

REVIEWERS' COMMENTS:

Reviewer #1 (Remarks to the Author):

The reviewers have addressed most of my point.

I only have a few comments, where the first is more of points for consideration than implementation. While I think the second should be addressed.

Point 2: orientation preference and attentional modulation. While the distributions show no clear structure, I would argue, in line with my previous point, that there are small hints of A being less often at sample orientation -90 deg, and it being more often at sample orientation +/- ~30 deg, i.e. on the steep flanks of tuning curves, where best discriminability exists. Bu I agree, that these distributions are inconclusive either way.

Solists-choristers: The argument does not make sense in that frame work. If the chorister couples to the soloist, then the soloist will not be identified as a soloist in the analysis of Okun et al. Both would be jointly modulated in the population activity, even if the second follows with a small delay.

Reviewer #2 (Remarks to the Author):

The authors did a thorough revision and addressed my concerns. I have no further comments.

Reviewer #3 (Remarks to the Author):

I appreciate the authors attention to my concerns - I have no reservations concerning the publication of this report in Nature Communications

REVIEWERS' COMMENTS:

Reviewer #1 (Remarks to the Author):

The reviewers have addressed most of my point.

I only have a few comments, where the first is more of points for consideration than implementation. While I think the second should be addressed.

Point 2: orientation preference and attentional modulation. While the distributions show no clear structure, I would argue, in line with my previous point, that there are small hints of A being less often at sample orientation -90 deg, and it being more often at sample orientation +/- ~30 deg, i.e. on the steep flanks of tuning curves, where best discriminability exists. But I agree, that these distributions are inconclusive either way.

Hm. It does sort of look like that. Certainly something interesting to keep an eye on during future work. Thanks for the suggestion.

Soloists-choristers: The argument does not make sense in that framework. If the chorister couples to the soloist, then the soloist will not be identified as a soloist in the analysis of Okun et al. Both would be jointly modulated in the population activity, even if the second follows with a small delay.

We see the reviewer's point. The reference to "soloists" and "choristers" was meant as shorthand to refer to the general property of diverse functional connectivity in a neural population, of which the framework described by Okun et al. is one example. The reference to "soloists" and "choristers" does not appear anywhere in our manuscript or supplemental materials, only in the response to reviews. Our argument was merely that diverse connectivity, as in our minimal example, but perhaps also manifested in other ways, can generate the pattern of results that we observed.

Reviewer #2 (Remarks to the Author):

The authors did a thorough revision and addressed my concerns. I have no further comments.

Thank you for your help and feedback.

Reviewer #3 (Remarks to the Author):

I appreciate the authors attention to my concerns - I have no reservations concerning the publication of this report in Nature Communications

Thank you, also, for your help and feedback.